

# Physicochemical characterization of free troposphere and marine boundary layer ice-nucleating particles collected by aircraft in the eastern North Atlantic

Daniel A. Knopf[1], Peiwen Wang[1], Benny Wong[1], Jay M. Tomlin[2], Daniel P. Veghte[3†], Nurun N. Lata[3], Swarup China[3], Alexander Laskin[2], Ryan C. Moffet[4], Josephine Y. Aller[1], Matthew A. Marcus[5], Jian Wang[6]

[1]School of Marine and Atmospheric Sciences, Stony Brook University, Stony Brook, NY 11794, USA
[2]Department of Chemistry, Purdue University, West Lafayette, IN 47907, USA
[3]Environmental Molecular Sciences Laboratory/Pacific Northwest National Laboratory, Richland, WA 99354, USA
[4]Sonoma Technology, Inc., Petaluma, CA 94954, USA
[5]Advanced Light Source, Lawrence Berkeley National Laboratory, Berkeley, CA 94720, USA
[6]Center for Aerosol Science and Engineering, Department of Energy, Environmental and Chemical Engineering, Washington University in St. Louis, St. Louis, MO 63130, USA

†Current address: Center for Electron Microscopy and Analysis, The Ohio State University, Columbus, OH 43212, USA

*Correspondence to*: Daniel A. Knopf (daniel.knopf@stonybrook.edu)

## Abstract

Atmospheric ice nucleation impacts the hydrological cycle and climate by modifying the radiative properties of clouds. To improve our predictive understanding of ice formation, ambient ice-nucleating particles (INPs) need to be collected and characterized. Measurements of INPs at lower latitudes in a remote marine region are scarce. The Aerosol and Cloud Experiments in the Eastern North Atlantic (ACE-ENA) campaign, in the region of the Azores Islands, provided the opportunity to collect particles in the marine boundary layer (MBL) and free troposphere (FT) by aircraft during the campaign's summer and winter intensive operation period (IOP). The particle population in samples collected was examined by scanning transmission X-ray microscopy with near-edge X-ray absorption fine structure spectroscopy. The identified INPs were analyzed by scanning electron microscopy with energy-dispersive X-ray analysis. We observed differences in the particle population characteristics in terms of particle diversity, mixing state, and organic volume fraction between seasons, mostly due to dry intrusion events during winter, and between the sampling locations of the MBL and FT. These differences are also reflected in the temperature and humidity conditions under which water uptake, immersion freezing (IMF), and deposition ice nucleation (DIN) proceed. Identified INPs reflect typical particle types within the particle population on the samples and include sea salt, sea salt with sulfates, and mineral dust, all associated with organic matter, and carbonaceous particles. IMF and DIN kinetics are analyzed with respect to heterogeneous ice nucleation rate coefficients, $J_{\text{het}}$, and ice nucleation active site density, $n_s$, as a function of the water criterion $\Delta a_w$. DIN is also analyzed in terms of contact angles following classical nucleation theory. Derived MBL IMF kinetics agree with previous ACE-ENA ground site INP measurements. FT particle samples show greater ice nucleation propensity compared to MBL particle samples. This study emphasizes that the types of



INPs can vary seasonally and with altitude depending on sampling location, thereby showing different ice nucleation propensities, crucial information when representing mixed-phase cloud and cirrus cloud microphysics in models.

**Short Summary (500 character in total)**

Ambient particle populations and associated ice-nucleating particles (INPs) were examined from particle samples collected
onboard aircraft in the marine boundary layer and free troposphere in the Eastern North Atlantic during summer- and wintertime. Chemical imaging shows distinct differences in the particle populations seasonally and with sampling altitudes which are reflected in the INP types. Freezing parameterizations are derived for implementation in cloud-resolving and climate models.

**Introduction**

Ice formation by atmospheric particles affects cloud formation, cloud albedo, and precipitation and, thus, the global radiative budget and the hydrological cycle (Boucher et al., 2013; Storelvmo, 2017; Mülmenstädt et al., 2015; McCoy et al., 2022; Mülmenstädt et al., 2021; McCoy et al., 2020; Murray and Liu, 2022). Supercooled droplets can freeze spontaneously via homogeneous nucleation when temperatures are below ~−38 °C while ice-nucleating particles (INPs) can initiate heterogeneous ice nucleation at higher temperatures and under saturated and subsaturated conditions (Pruppacher and Klett,
1997; Cantrell and Heymsfield, 2005; Knopf et al., 2018; Vali et al., 2015; Kanji et al., 2017; Knopf and Alpert, 2023). Although atmospheric INPs are scarce, typically on the order of 1 in ~$10^5$ ambient particles in a liter of air in the troposphere (DeMott et al., 2010), INPs play an central role in cloud microphysical processes (Peter et al., 2006; Baker, 1997; Storelvmo, 2017; Murray and Liu, 2022). Nevertheless, our predictive understanding of atmospheric particles which act as INPs under conditions typical of mixed-phase clouds in which ice crystals and supercooled liquid droplets coexist and cirrus clouds where
only ice crystals exist, is still insufficient, and, thus, the implementation of INP predictions in cloud and climate models remains challenging (Boucher et al., 2013; Storelvmo, 2017; Cesana and Del Genio, 2021; McCoy et al., 2016; Murray and Liu, 2022). One possible explanation for this is the insufficient description of the particles' ice nucleating properties and knowledge of dominant types of INPs in various atmospheric environments (Kanji et al., 2017; Knopf et al., 2018; Cziczo et al., 2017; Knopf and Alpert, 2023).

60       The Eastern North Atlantic (ENA) is a region of complex aerosol sources and diverse aerosol composition (Wood et al., 2015; Zheng et al., 2018). Aerosol sources which contribute to the ENA aerosol include natural emissions from the ocean, anthropogenic emissions from North America, South American and African biomass burning regions, and high mineral dust-laden air masses from the Sahara (Hamilton et al., 2014; Zheng et al., 2020; Alonso-Perez et al., 2012). From June 2017 to February 2018, the Aerosol and Cloud Experiments in the Eastern North Atlantic (ACE-ENA) campaign consisting of airborne
deployments and a stationary Atmospheric Radiation Measurement Climate Research Facility was conducted on Graciosa Island in the Azores (Knopf et al., 2022; Wang et al., 2022; Zawadowicz et al., 2021; Tomlin et al., 2021; Wang et al., 2021).



The overarching goals of the ACE-ENA campaign were to advance our understanding of marine MBL clouds through field measurements of cloud condensation nuclei (CCN), drizzle, and cloud microphysics, to close knowledge gaps in aerosol and cloud processes, and to validate and improve retrieval algorithms of surface-based remote sensing (Wang et al., 2022).

Although the focus of this campaign was on liquid cloud processes, it provided a unique opportunity to examine airborne-collected particles in the marine MBL and FT as potential sources of INPs in a remote region that is also impacted by aerosol sources far away. During wintertime this region also experiences dry intrusion events (DI) from the FT into the marine MBL producing deepening, drying, and cooling conditions (Ilotoviz et al., 2021; Raveh-Rubin, 2017), which impact the composition of the MBL particle population (Tomlin et al., 2021).The acquired airborne measurements also allowed comparison to

previously conducted ground-site INP measurements in this region (Knopf et al., 2022; China et al., 2017; Lata et al., 2021).

In the marine MBL of ENA, sea spray aerosol (SSA) dominates in the large mode particles, while entrained FT particles from anthropogenic sources contribute to most of the Aitken and accumulation mode particles (Zheng et al., 2018), the latter  impacting cloud microphysical properties (Wang et al., 2020). It has been observed that long-range transported and aged wildfire aerosol in the ENA marine MBL can serve as CCN and, thus, impact local cloud formation (Zheng et al., 2020).

During the ACE-ENA campaign elevated CCN number concentrations in the FT were observed that did not impact the MBL CCN pool significantly (Wang et al., 2021). However, entrainment of particles too small to act as CCN from the FT into the MBL can act as CCN after condensational growth. Wang and colleagues further concluded that surface measurements overestimate the number of CCN relevant to the formation of MBL clouds under decoupled conditions (Wang et al., 2021). *Tomlin et al.* (2021) employed single-particle micro-spectroscopic techniques to examine the external and internal mixing state

for airborne-collected particle populations sampled during the ACE-ENA campaign. They found that carbonaceous particles were the dominant type in this region, while DI events significantly impacted the particle population by changing the carbonaceous and inorganic contributions. DI events resulted in a decreased contribution of organic compounds to the population in the MMBL and FT while the contribution of inorganic species increased. This in turn has a direct effect on the particles' hygroscopicity and thus CCN properties leading *Tomlin et al.* (2021) to conclude that entrainment of particles from

long-range continental sources alters the mixing state of the particle population and the CCN properties of aerosol particles in this region.

Less information about INPs in the ENA region is available. However, since ACE-ENA was conducted in a remote marine region, nascent or aged SSA is well known to have associated organic matter (OM) (Pham et al., 2017; Aller et al., 2017; Facchini and O'Dowd, 2009; O'Dowd et al., 2004; Cochran et al., 2017; Lee et al., 2020; Ault et al., 2013a; Alpert et al.,

2015) which can serve as INPs (Cornwell et al., 2021; McCluskey et al., 2018; McCluskey et al., 2017; DeMott et al., 2016; Alpert et al., 2022; Ladino et al., 2016; Wilson et al., 2015; Knopf et al., 2011; Alpert et al., 2011b; Alpert et al., 2011a; Schnell and Vali, 1976; Schnell, 1975; Wilbourn et al., 2020; Irish et al., 2019; Creamean et al., 2019; Cornwell et al., 2019) and thus would be  expected to contribute to the regional INP pool (Lata et al., 2021; Knopf et al., 2022; China et al., 2017). Other sources of particles which might contribute to the INP population include long-range transported carbonaceous particles and

dust particles originating from the north African deserts as well as primary organic aerosol particles from, e.g., fossil fuel





combustion and biomass burning (Knopf et al., 2018; Kanji et al., 2017). Secondary organic aerosol (SOA) originating from anthropogenic and biogenic precursor gases have been shown to initiate ice formation (Kanji et al., 2017; Wang et al., 2012a; Wolf et al., 2020; Knopf et al., 2018). Finally, while inorganic mineral dust particles are well known to be some of the most efficient inorganic INPs (e.g., Kanji et al., 2017; Cziczo et al., 2017; Murray et al., 2012), OM associated with soil dust particles

can also serve as potent INPs (Knopf et al., 2021; Creamean et al., 2020; Hill et al., 2016; Tobo et al., 2014).

*China et al.*, (2017) report on the examination of particles collected at the Observatory of Mountain Pico 2225 m above sea level located on a neighboring Azores Island, which were studied for their propensity to initiate immersion freezing (IMF) and deposition ice nucleation (DIN). IMF describes the process of ice nucleation from an INP immersed in a supercooled liquid droplet while DIN represents the ice formation pathway where ice forms from the supersaturated vapor phase directly

on the INP (Knopf et al., 2018; Vali et al., 2015). Though recent studies show that DIN could be the result of pore condensation freezing, where homogeneous freezing in nanometer-sized pores is facilitated at lower relative humidity with respect to ice (RH$_{ice}$) than needed for homogeneous ice nucleation (David et al., 2019; Marcolli, 2014, 2020). *China et al.* (2017) demonstrated that most particles were coated by OM and that the identified INPs contained mixtures of dust, aged sea salt and soot, and OM acquired either at the source or during transport and that ice formation was promoted by both IMF and DIN

pathways (China et al., 2017). These findings have been corroborated in a recent study that also examined particle samples collected at the Observatory of Mountain Pico using micro-spectroscopic analyses of the ambient particle population and INPs (Lata et al., 2021). In this study, IMF and DIN measurements were related to the size-resolved chemical composition, mixing state (MS), distribution of OM, functional groups, and phase state of the ambient particle population. In the case of identified and characterized INPs, particulate OM, sulfate content, and phase state were found to influence the properties of INPs (Lata

et al., 2021). Specifically, highly viscous particles showed a greater propensity to initiate DIN at temperatures below 220 K compared to less viscous (liquid-like) particles. To relate the physicochemical properties of the particle population with the identified INPs, *Knopf et al.* (2022) analyzed particle samples collected at the ACE-ENA ground site during summer 2017 to relate the physicochemical properties of the particle population with the identified INPs. The identified INP types were found to consist of fresh sea salt with organic compounds or processed sea salt containing dust and sulfur with organic compounds

and showed the same organic spectral features as SSA INPs (Alpert et al., 2022; Knopf et al., 2022). That recent study supports the contention put forth by *Knopf et al.*(2022), that the MBL can serve as a source of IMF and DIN INPs.

Results reported in this study extend previous INP measurements of ambient particles collected during ACE-ENA (Knopf et al., 2022) examining particle samples collected in the MBL and FT regions by aircraft during summer and winter and in the presence and absence of DI events. We analyzed 7 particle samples collected during 6 different flights including 3

samples from the MBL and 4 from the FT which allows for examination of the role of dynamics (transport and DI events), sampling location, and season in determining the physicochemical properties of the aerosol population and INPs. Additionally, analysis of these aircraft collected samples will be compared to ground site aerosol and INP measurements conducted during the same field campaign (Knopf et al., 2022). Scanning transmission X-ray microscopy with near-edge X-ray absorption fine structure spectroscopy (STXM/NEXAFS) is used to determine the organic speciation, MS, and organic volume fraction (OVF)



of the individual particles thereby characterizing the particle population in each sample. Scanning electron microscopy with energy dispersive X-ray analysis (SEM/EDX) is used to examine the morphology and elemental composition of the identified INPs which are then placed in context with the investigated particle population. The thermodynamic conditions, i.e., temperature (T) and $RH_{ice}$, at which collected particles induce IMF and DIN are assessed. The ice nucleation kinetics of IMF and DIN kinetics will be assessed using the classical nucleation theory (CNT) rooted water-activity based immersion freezing

model (ABIFM) (Knopf and Alpert, 2013) and the ice nucleation active sites (INAS) approach (Vali, 1971; Connolly et al., 2009). The acquired ice nucleation kinetics will be compared to IMF and DIN kinetics descriptions made on particle samples collected at the ACE-ENA ground site during summer 2017 (Knopf et al., 2022). These are applied to derive formulations of corresponding IMF and DIN parameterizations applicable to MBL and FT aerosol of the ENA region.

## 2. Experimental methods

**2.1 Particle collection**

        Particle sampling for the ACE-ENA campaign has been described in detail in *Tomlin et al.* (2021) and, thus, is only briefly summarized here. Particles were collected aboard the U.S. Department of Energy Gulfstream aircraft (G-1) by impaction on to various substrates. Typical flight patterns were flown between Terceira Island and Graciosa Island, Portugal, and within 20–30 km of Graciosa Island (Tomlin et al., 2021; Wang et al., 2022). The aerosol was provided at 1 SLPM

(standard liter per minute) by an isokinetic inlet installed on the G-1 aircraft to a time-resolved aerosol collector (TRAC) a single-stage impactor sealed against ambient conditions with an aerodynamic cut-off diameter, $D_{50\%} = 0.36\ \mu m$. The TRAC houses a substrate disc that holds up to 160 substrates for collection of particles via impaction at preset time intervals (Tomlin et al., 2021; Laskin et al., 2006; Laskin et al., 2003). The collected particle sizes typically varied between 0.1 and 5 µm. After all substrates are used, the particle samples were removed under particle-free conditions in the laboratory and stored in

transmission electron microscopy (TEM) grid boxes in a low humidity storage cabinet (Knopf et al., 2022; Knopf et al., 2021; Tomlin et al., 2021; Knopf et al., 2014).

        In this study, we employed $Si_3N_4$-coated silicon wafer chips (Silson Ltd.) for ice nucleation experiments and TEM grids (copper 400 mesh grids, carbon type-B film, Ted Pella, Inc.) for SEM/EDX and STXM/NEXAFS analyses (Knopf et al., 2022; Knopf et al., 2021; Charnawskas et al., 2017).

Table 1 provides sampling date, time, and altitude of the particle samples discussed in this study. IOP1 and IOP2 indicate the intensive operation periods during the summer and winter, respectively. TRAC impacts particles onto only one substrate for each collection time interval. Hence, collection of particles onto different substrates for examination using different analytical techniques at the same time is not possible. Therefore, substrates for INP measurements and for particle composition do not exactly reflect the same air mass. The difference in sampling heights and collection times are given in

Table 1. In some instances, STXM/NEXAFS measurements performed on two TEM grids were combined. Collection periods that were impacted by DI events are indicated in Table 1. DI events are defined by air masses that experience a pressure





increase of 400 hPa within a 24 h window within 3° radius around Graciosa Island (Tomlin et al., 2021; Ilotoviz et al., 2021; Raveh-Rubin, 2017). DI events in the ENA region are most common during the winter times. The examined summertime collected (IOP1) samples were not affected by DI events. Figures S1 and S2 present ensemble 10-days HYSPLIT backward

trajectory calculations at the sample location during flight (Stein et al., 2015). During summer MBL measurements (IOP1-MBL1), the air masses resided in the MBL for most of the backward trajectory analysis. However, during wintertime MBL sampling flights (IOP2-MBL1 and 2), the trajectories indicate that air masses also originated from the FT. These trends coincide with the identified DI events. For summer FT sampling (IOP1-FT1), the trajectories indicate that the air masses resided for a couple of days at similar altitudes and may have been influenced by MBL air masses after longer times. For

wintertime FT sampling (IOP2-FT1-3), air masses resided for a few days at similar altitudes but at later times originated from higher altitudes. Qualitatively, there seem to be different trends in the air mass trajectories for sampling during summer and winter times.

## 2.2 Single-particle analysis by X-ray and electron microscopy

STXM/NEXAFS (Kilcoyne et al., 2003) which allows speciation of the particulate organic carbon (OC), and

assessment of the particulate OVF was employed to examine the MS of the collected particle population. This microscope was operated at the carbon K-edge (278–320 eV) to enable physicochemical characterization of the particle composition at a spatial resolution of ~35 nm following our previous studies (Knopf et al., 2022; Knopf, 2023; Knopf et al., 2021; Tomlin et al., 2021; Knopf et al., 2014; Laskin et al., 2019; Laskin et al., 2016; Moffet et al., 2013; Wang et al., 2012b; Moffet et al., 2010; Knopf et al., 2010; Fraund et al., 2020). Since we compare ACE-ENA airborne particle samples with ground site samples published

recently (Knopf et al., 2022), STXM/NEXAFS measurements and analysis follow the procedures outlined in this previous study (Knopf et al., 2022). We performed two types of analyses to either examine a larger number of particles or acquire a more detailed analysis of fewer particles. STXM images were recorded at four selected X-ray energies at 35 nm resolution and 1 ms dwell time including 278 eV (pre-edge, inorganic: IN), 285.4 eV (C=C, elemental carbon: EC), 288.5 eV (COOH: OC), and 320 eV (post-edge). This analytical procedure allows imagining of the chemical composition and determination of the MS

and OVF of the individual particles (Knopf et al., 2022; Moffet et al., 2010; Hopkins et al., 2007; Fraund et al., 2020). We used NaCl and adipic acid species as representative mass absorption coefficients for the IN and OC particle fractions, respectively (see also discussion in *Fraund et al.*, 2019). The accuracy of the representation of the particle population depends on the number of particles analyzed. For example, if the aim is to have the probability to be at least 0.95 that all estimates are within 0.1 of the population particle types, a sufficient sample size of randomly chosen particles is 403 particles (Thompson,

1987). Sample sizes for STXM analysis varied between 71 to 352 particles, resulting in particle population uncertainties ranging from greater than 50% to less than 10%. In other words, if, e.g., 30% of the particles are identified to be organic carbon-inorganic mixtures (OCIN), the uncertainty in derived representativeness will be ±>15 and ±<3%, respectively. In addition, for selected particles high-resolution energy absorption (NEXAFS) spectra examining 96 energies at 35 nm pixel



resolution and 1 ms dwell time were recorded allowing for a detailed interpretation of the particulate OC including organic
functionalities.

Particles that were identified as INPs in our ice nucleation experiments by optical microscopy, were individually examined by SEM/EDX for morphological features and elemental composition in a similar fashion as in our previous studies (Knopf et al., 2022; China et al., 2017; Knopf et al., 2014). The SEM/EDX was operated at 20 kV (FEI Quanta 3D, EDAX Genesis). Besides recording electron micrographs for imaging purposes, analysis of the EDX spectra allows quantification of
the relative atomic fractions of the following elements: C, N, O, Na, Mg, Al, Si, P, S, Cl, K, Ca, Mn, and Fe.

For analysis of the ice nucleation kinetics, accurate particle surface area is needed. In addition to particle surface area estimates based on optical microscopy, we used particle surface area estimates obtained by computer-controlled SEM/EDX (CCSEM/EDX) (Tomlin et al., 2021) due to its superior resolution compared to optical microscopy (Table 2). When available, these data were acquired from samples collected during the same flights at similar altitudes as the particle samples investigated
for this study (Table 1). For each individually analyzed particle, CCSEM/EDX provides an area equivalent diameter (AED) which represents a circle with a diameter that has the same surface area as the imaged particle. The AED is used to estimate the total particle surface area present in the ice nucleation experiments by assuming the particles to be half spheres. As discussed in *Knopf et al.* (2022), underestimation of the actual particle surface area due the presence of nanoscale surface features would imply that derived ice nucleation kinetics represent upper limits.


**2.3 Optical microscopy-based ice nucleation experiment**

Particles collected on the $Si_3N_4$ coated Si wafer chip substrates were investigated for their ability to serve as INPs at given T and $RH_{ice}$. A custom-built ice nucleation cell which includes a cryo-cooling stage coupled to an optical microscope was used to examine IMF and DIN for temperatures as low as 200 K and up to water saturation (Alpert et al., 2022; China et
al., 2017; Charnawskas et al., 2017; Wang et al., 2012b; Knopf et al., 2011). This setup allows the temperature-controlled particles on the substrate to be exposed to different water partial pressures which are controlled by a humidified flow of dry nitrogen (ultra-high purity) at 1 standard liter per minute. As the gas passes through and exits the nucleation cell, its dew point is determined using a chilled mirror hygrometer. Substrate temperature and dew point yield $RH_{ice}$. Water uptake and ice formation can be identified by changes in particle size and morphology when the changes are greater than 1 and 0.2 µm in size
at magnifications of 230✕ and 1130✕, respectively (Wang and Knopf, 2011). The dew point temperature uncertainty is ± 0.15 K. The uncertainties for $RH_{ice}$ depend on the probed temperature region and range from about ± 4.0 to ± 5.7 % where the uncertainty increases for lower substrate temperatures. The substrate temperature is calibrated by measuring the melting points of different organic species and ice (Rigg et al., 2013). In addition, controlled ice crystal growth and sublimation experiments are conducted to verify the substrate temperature uncertainty (Wang and Knopf, 2011).
A typical experiment starts at subsaturated conditions, i.e., $RH_{ice} < 100\%$. Then $RH_{ice}$ is increased by ~ 2 % min$^{-1}$, reflecting typical cirrus cloud formation conditions, and water uptake and ice formation are recorded. Every 0.02 K, or 12 s, an image is taken providing time, substrate temperature, and dew point. This information is used to identify IMF and DIN



modes and to derive heterogeneous ice nucleation rate coefficients, $J_{het}$, in units cm$^{-2}$ s$^{-1}$ and INAS density, $n_s$, in units cm$^{-2}$ following analyses described in detail previously (Alpert et al., 2022; Knopf et al., 2022; Knopf et al., 2020; Alpert et al.,
2011a; Wang and Knopf, 2011; China et al., 2017). Only the first observed ice nucleation event is applied in the ice nucleation kinetics analysis since subsequent ice crystal formation can lead to an inhomogeneous field of humidity, rendering the RH$_{ice}$ value at which ice formed uncertain (Knopf, 2023).

Digital imaging analysis allows to relocation of the identified INP on the substrate in other micro-spectroscopic analytical instruments such as SEM/EDX (Knopf et al., 2022; Alpert et al., 2022; Knopf et al., 2014). Once ice formation is
detected, ice sublimation under higher resolution (1130✕) is performed as shown in Fig. S3a and b. The residual particle is identified as the INP. A sequence of optical microscope images showing different field of views are used to relocate the INP in other microscopic instruments by means of pattern recognition and triangulation. Figure S3c and d show a magnified field of view and the identified INP, respectively, using SEM.

## 3. Results and discussion

### 3.1 Particle population characterization

Figure 1 provides representative false-color particle population MS images and size-resolved MS and OVF analyses for MBL particle samples derived from STXM/NEXAFS analyses. IOP1-MBL1 sample, collected during summer, shows a different particle population MS compared to the wintertime IOP-MBL1 and MBL2 samples. Additional particle population MS images are shown in Fig. S4 that corroborate this trend. Particles in the IOP1-MBL1 sample are smaller and dominated by
OCIN particles with few OC particles. IOP2-MBL1 sample indicates the presence of OCIN particles with thick coatings of OC. This is also the case for sample IOP2-MBL2, though particles seem to be larger, contain C=C double bonds, and generally appear to be chemically more complex. Size-resolved MS and OVF analyses for sample IOP1-MBL1 indicate that OCIN particles dominate the population, with a significant fraction of IN particles in the smallest size range, and the majority of particles having an OVF between 20 and 60 %. The particle size distribution is unimodal with the largest particles smaller than
1 μm. MBL samples from IOP2 (collected during winter) show a bimodal particle size distribution including particles with sizes up to 2 μm. Sample IOP2-MBL1 is dominated by OCIN particles with OVFs between 20 to 60% while particles in sample IOP2-MBL2 are more diverse and include EC and OC particles resulting in OVF ranging from 20 to 80 %.

Figure 1 and Fig. S4 demonstrate a difference in particle population MS between IOP1 and IOP2 MBL samples. Seasonal change from summer to winter affects ocean conditions and weather patterns that can result in DI events. Figures S5
and S6 show Aqua MODIS (Moderate Resolution Imaging Spectroradiometer) derived monthly mean sea surface temperatures and chlorophyll $a$ concentrations (Nasa Goddard Space Flight Center, 2019, 2022). During summer, ocean water in the ENA region is warmer and it has less biological production. During wintertime chlorophyll $a$ concentrations in waters surrounding the Azores Islands are greater (Fig. S6). The ENA region likely also experiences greater wind speeds and more sustained winds during the winter generating more and larger SSA particles. These factors could impact primary marine organic emissions



associated with SSA particles. Also, we have shown that during wintertime the Azores region is prone to DI events (Tomlin et al., 2021). As shown in Table 1, IOP2-MBL1 and MBL2 samples were collected as Graciosa Island region experienced DI events. These DI events can significantly change the characteristics of the MBL particle population resulting in the presence of particles with complex composition (Tomlin et al., 2021).

Representative false-color particle population MS images and size-resolved MS and OVF analyses for FT particle

samples derived from STXM/NEXAFS analyses are shown in Figure 2. Images from sample IOP1-FT1 display numerous submicron-sized particles dominated by OCIN particles with EC and IN particles contributing to those in the smaller sizes. The corresponding OVF ranges between 20 and 60 %. As in the case of the MBL samples, the IOP1-FT1 sample collected during summer shows significantly different population characteristics compared to the wintertime collected FT samples. These differences are demonstrated further in Fig. S7. Backward trajectories suggest that air masses associated with IOP1-FT0

remained for the most part over the Atlantic Ocean (Fig. S1) while IOP2-FT1 and FT2 might have been impacted by air masses passing over the eastern continental north America (Fig. S2). Particles collected in sample IOP2-FT3 appear to have originated solely from greater altitudes (Fig. S2). Particles in sample IOP2-FT1-3 show greater MS diversity, include many in the larger size range, and the population includes particles with greater OVFs compared to sample IOP1-FT1. OCIN particles still represent a major particle-type fraction but, specifically, for samples IOP2-FT2 and FT3, EC, OC, and IN particles, and

mixtures thereof, also contribute significantly to the particle population. The sample IOP2-FT3, collected at the highest altitude of about 4056 m (Table 1), exhibits the highest OVFs, characterized by the largest contribution of purely OC particles compared to other samples. IOP2-FT1 and FT3 samples experienced DI events that potentially impacted the particle population. Based on larger particle samples and application of CCSEM/EDX, *Tomlin et al.* (2021) observed that the carbonaceous contributions to the FT particle population decreases while the inorganic contribution increases during DI events.

The FT samples are expected to be impacted by long-range transport, thus, physicochemical transformations by aging processes are expected (Tomlin et al., 2021; China et al., 2017; Lata et al., 2021). Hence, transport dynamics and DI events are likely impacting the FT particle population to a greater extent than changes in seawater characteristics resulting from seasonal differences in the abundances and activities of planktonic microorganisms.

Figure 3 presents OVF false color images and high resolution NEXAFS spectra representative of MBL and FT particle

samples. The MBL samples are dominated by OCIN particle types which may reflect the island location and the fact that SSA particles from the marine environment consist of an inorganic core coated with varying  amounts of organic material (Alpert et al., 2022; Laskin et al., 2016; Knopf et al., 2014; Pham et al., 2017; Ault et al., 2013b; Lee et al., 2020; Cochran et al., 2017). The NEXAFS spectra of the organic particle fraction presented in Fig. 3 shows the typical signatures of fresh and aged sea salt particles including the absorption bands of carboxyl and carbonate groups and the presence of potassium (Knopf et al., 2014;

Alpert et al., 2022; Laskin et al., 2012; Knopf et al., 2022; Pham et al., 2017). In fact, the organic coating displays the same NEXAFS spectral features as SSA particles that have been demonstrated to serve as INPs (Alpert et al., 2022). As discussed in *Alpert et al.* (2022) the carbonate signal is due to beam damage with the conversion of the carboxyl group into a carbonate





group. Detailed particle images in Fig. 3 show the distribution of identified particle species where OC, carbonate, and potassium are located on the outside of the particle.

In Fig. 3 it is apparent that particles collected in the FT have greater MS diversity, consistent with different NEXAFS spectral features. Included are particles dominated by elemental carbon signatures, aged sea salt particles, mineral rich particles, and particle mixtures. Corresponding particle images display greater physicochemical complexity compared to MBL particles. These findings agree with the observations by *Tomlin et al.* (2021).

### 3.1 INP identification

Twelve INPs from the FT samples were identified and analyzed for their morphology and elemental composition using SEM/EDX. Figure 4 shows electron micrographs of INPs active in the DIN mode as well as associated observed ice formation temperatures. These INPs are categorized as sea salt, sea salt + sulfate, and OC particles based on their elemental composition (Table 3). The INP elemental compositions derived by SEM/EDX and corresponding EDX spectra are summarized in Fig. 5. Figures S8 and S9 provide more detailed EDX spectra and additional INP images. The INPs identified
for these three different FT samples vary in composition (Table 3), although all of them have OM associated with them. Identified INPs from IOP1-FT1 are comprised mostly of sea salt particles, though mineral dust and one unidentified particle type were also observed to act as INPs. INPs associated with IOP2-FT1 display features of sea salt in the presences of sulfates, hinting at processed or aged particles. INPs from IOP2-FT3 samples are purely carbonaceous in nature. No traces of other elements could be detected in those two INPs (Figs. 5 and S9). In this case, we can only speculate that these organic particles
were likely in a glassy phase state when acting as INPs at -50 °C.

The FT INPs identified here are compared to INPs characterized from ground collected samples during the ACE-ENA campaign (Knopf et al., 2022). That study also identified sea salt, processed sea salt, and carbonaceous particles as INPs. INP measurements and identification at the Observatory of Mountain Pico site also observed aged sea salt, carbonaceous, and sulfate and carbonaceous coated dust particles which served as INPs (China et al., 2017; Lata et al., 2021). Overall, we find
that the majority of identified INPs on all samples regardless of where collected belong to the major particle-types of the particle population and are not different or unique particles. This finding is consistent with similar findings from other field campaigns that have applied micro-spectroscopic analytical techniques to examine particle populations and identified INPs (Hiranuma et al., 2013; Knopf et al., 2022; China et al., 2017; Knopf et al., 2014; Lata et al., 2021).

### 3.2 Ice nucleation experiments

The ice nucleation experiments for the 7 particle samples are summarized in Fig. 6. Observed water uptake, IMF, and DIN events are plotted as a function of particle temperature and $RH_{ice}$. The IOP1-MBL1 sample shows different ice nucleation propensity compared to the IOP2-MBL1 and MBL2 samples that were collected during DI events. Water uptake occurs at lower $RH_{ice}$ compared to the IOP2-MBL samples. Only for the IOP1-MBL1 sample, IMF was observed at about 240 K below and at saturation indicating greater IMF propensity compared to IOP2-MBL samples. Also, IMF proceeded over a greater




range of RH$_{ice}$ and at lower RH$_{ice}$ compared to IOP2-MBL samples. Lastly, DIN proceeds, on average, at lower RH$_{ice}$ compared
to IOP2-MBL samples. Overall, these ice nucleation results reflect the above discussed differences in particle populations
between the IOP1-MBL1 and IOP2-MBL samples. Furthermore, it highlights the importance of DI events on MBL particle
populations as discussed in *Tomlin et al.* (2021) and, potentially, on INP sources.

Since the INPs are organic particles and/or are associated with OC coatings, Fig. 6 includes the glass transition
temperatures, T$_g$, of several aerosol types for comparison, including SOA particles formed from the precursor gases α-pinene,
naphthalene (Charnawskas et al., 2017) and ambient SOA and Suwannee river fulvic acid particles (Wang et al., 2012a). Also
included is the full deliquescence relative humidity of SOA generated from α-pinene precursor gas, valid for the experimentally
applied humidification rate (Charnawskas et al., 2017). DIN can proceed on glassy organic particles while IMF can proceed
until the full deliquescence relative humidity is reached (Knopf et al., 2018). Once the particle is fully deliquesced, only
homogeneous freezing can proceed when temperatures are sufficiently low (Berkemeier et al., 2014; Zobrist et al., 2008). Of
these particle types, fulvic acid has the highest T$_g$ (Fig. 6), likely because it is a large macromolecule (Koop et al., 2011). SSA-
INPs consist of polysaccharidic and proteinaceous macromolecular compounds (Alpert et al., 2022), which likely possess a
high T$_g$ (Koop et al., 2011; Shiraiwa et al., 2011). DIN for MBL samples falls in the range of glassy fulvic acid particles. One
can expect the full deliquescence relative humidity of fulvic acid to be significantly higher than for α-pinene derived SOA,
possibly explaining the observed IMF for temperatures as high as 240 K in the case of IOP1-MBL1 sample. This may point to
the importance of glassy carbonaceous particles or particles coated by glassy OM to serve as INPs, although further
microscopic investigations are necessary to confirm such a pathway. Figure 6 further includes the range of pore condensation
freezing for pore sizes of 7.5 to 20 nm (Marcolli, 2020, 2014). Observed DIN events for IOP1-MBL1 samples fall in this range
of pore condensation freezing, however, for IOP2-MBL1 and MBL2 observed ice formation falls mostly outside the pore
condensation freezing active range of temperature and RH$_{ice}$.

As pointed out above, the STXM derived particle morphology and NEXAFS spectral features of the organic
compounds associated with the particles in the MBL samples, are identical to the OM detected in SSA particles and SSA-INPs
(Alpert et al., 2022). One would expect particles having similar physicochemical properties, similar ice nucleation
characteristics with respect to the measured different nucleation pathways and thermodynamic conditions should be observed.
Indeed, the range of water uptake, IMF, and DIN for IOP1-MBL1 in Fig. 6 (not affected by DI events, but dominated by SSA
emissions) match the conditions of SSA-INPs associated with various mesocosm experiments with marine microorganisms
and field-collected particles (*Alpert et al.* (2022), their Fig. 1). These results further support the conclusions by *Alpert et al.*
(2022) that "all SSA particles derived from aquatic environments regardless of the specific makeup of the planktonic
community can serve as INPs because they contain the same ice-nucleating agents". It also demonstrates that instrumentation
that can resolve physicochemical properties on the nanoscale, allow for predictive understanding of ice nucleation.

Compared to the MBL samples, FT samples show water uptake and DIN at lower RH$_{ice}$ (Fig. 6). Particles in sample
IOP-FT1 which were not affected by DI, and particles in IOP2-FT3 which were collected at 4000 m, the highest altitude, show
IMF at the highest temperatures of about 240 K. IMF for IOP2-FT2 and FT3 samples proceeds only at about 230 K. Overall,





FT samples demonstrate greater ice nucleation propensity compared to the MBL samples, while IOP1-FT1 shows the greatest ice nucleation efficiency among all samples.

Previous INP measurements at the Observatory of Mountain Pico that typically resides in the free troposphere (China et al., 2017; Lata et al., 2021) show DIN and IMF conditions similar to the FT samples investigated here. *Lata et al.* (2021) have shown that FT samples include solid organic particles that are likely responsible for DIN. The example $T_g$ and full deliquescence relative humidity curves shown in Fig. 6 suggest that DIN and IMF proceeded on highly viscous or solid 370 particles. Clearly, these collective observations emphasize the need for a better understanding of how highly viscous OC species interact with water initiating ice nucleation.

**3.2 Immersion freezing kinetics**

IMF rates are expressed by $J_{\text{het}}$ and $n_s$ (Knopf et al., 2022; China et al., 2017; Alpert et al., 2011a). We apply ABIFM (Knopf and Alpert, 2013) to parameterize $J_{\text{het}}$ and $n_s$ as a function of the water activity criterion, $\Delta a_w$, where $\Delta a_w(T) =$
$a_w(T) - a_w^i(T)$ (Koop et al., 2000). A necessary condition for the application of $\Delta a_w$ is that condensed-phase water activity, $a_w$, is in equilibrium with adjacent water partial pressure, resulting in $a_w =$ RH. $a_w^i(T)$ is the ice melting curve for frozen aqueous solutions as a function of water activity (Koop and Zobrist, 2009). Application of ABIFM allows the expression of heterogeneous freezing kinetics as a function of temperature and RH with a single parameter, thereby also covering the subsaturated temperature range (Knopf and Alpert, 2023). $\Delta a_w = 0$ represents the melting point of ice at 273.15 K and $\Delta a_w =$
0.313 represents the homogeneous ice nucleation temperature at 238 K and $J_{\text{hom}} \sim 10^{10}$ cm$^{-3}$ s$^{-1}$ (Koop et al., 2000).

Figure 7 shows derived IMF $J_{\text{het}}$ and $n_s$ values as a function of $\Delta a_w$, for MBL and FT samples. With increasing $\Delta a_w$ $J_{\text{het}}$ and $n_s$ are increasing. For MBL samples, average $J_{\text{het}} \sim 200$ cm$^{-2}$ s$^{-1}$ and $n_s \sim 2000$ cm$^{-2}$. A recent study by *Cornwell et al.* (2021) derived $J_{\text{het}}(243 \text{ K}) \sim 1000$ cm$^{-2}$ s$^{-1}$ corresponding to $\Delta a_w \sim 0.25$, about a factor of three greater than our derived $J_{\text{het}}$ values but still within the uncertainties of both measurements. The $n_s$ values at 243 K agree with previous laboratory and
field measurements of SSA and MBL field collected particles reported by *McCluskey et al.* (2017) and *DeMott et al.* (2016). FT samples show a trend of greater $J_{\text{het}}$ and $n_s$ values compared to MBL samples with average $J_{\text{het}} \sim 700$ cm$^{-2}$ s$^{-1}$ and $n_s \sim$ 8000 cm$^{-2}$. Though this trend is within the stated uncertainties of $J_{\text{het}}$ and $n_s$. For the MBL samples, $J_{\text{het}}$ and $n_s$ data are expressed by a linear regression following ABIFM, where $\log J_{\text{het}} = m a_w + c$ and $\log n_s = m a_w + c$, respectively and the fit parameters and associated uncertainties are given in Table 4. Since IMF FT data do not show a clear trend with $\Delta a_w$ no linear
regression is provided. A possible reason for the greater scatter in FT $J_{\text{het}}$ and $n_s$ may be the greater chemical diversity and complexity of the particles present in these samples.

When comparing derived IMF $J_{\text{het}}$ with $J_{\text{het}}$ for illite (Knopf and Alpert, 2013), natural mineral dust (Alpert and Knopf, 2016; Niemand et al., 2012), leonardite, a humic acid (Rigg et al., 2013; Knopf and Alpert, 2013), and diatomaceous material (Knopf and Alpert, 2013; Knopf et al., 2011; Alpert et al., 2011b; Alpert et al., 2011a), the field derived $J_{\text{het}}$ values
show a shallower slope, and therefore greater $J_{\text{het}}$ at lower $\Delta a_w$ (i.e., higher temperature) and lower $J_{\text{het}}$ at larger $\Delta a_w$ (i.e.,



lower temperature). In Fig. 7, we also included a recently derived ABIFM IMF $J_\text{het}$ parameterization for SSA which act as INPs (Alpert et al., 2022). Since the observed ice formation conditions and particle composition are similar to the SSA studied in *Alpert et al.* (2022), similar $J_\text{het}$ values are expected. However, this parameterization also displays a greater slope with $\Delta a_w$ compared to the field derived IMF $J_\text{het}$. The most likely explanation for this difference lies in the IMF analysis itself. *Alpert et*
*al.* (2022) analyzed the IMF data of several data sets with a stochastic freezing model that accounts for particle surface area uncertainties. As shown in *Alpert and Knopf* (2016), the inclusion of particle surface area uncertainty produces a steeper slope of IMF $J_\text{het}$, compared to the one when assuming that all particles possess the same surface area. This can amount to differences in $J_\text{het}$ at the lower and higher temperatures of several orders of magnitude (Alpert and Knopf, 2016). Accounting of particle surface uncertainty, however, has not been performed for these field derived data and are beyond the scope of this study.

The ABIFM IMF $J_\text{het}$ parameterization derived from ground site measurements during ACE-ENA (Knopf et al., 2022) agree with the IMF $J_\text{het}$ derived for MBL samples (Fig. 7). Although the ground measurements were based on samples with different particle loading and different INP sizes, the IMF kinetics are very similar when accounting for particle surface area. This finding supports the usefulness of ground site INP measurements in a well-mixed MBL. ABIFM IMF $J_\text{het}$ parameterization derived from particle samples collected at Observatory of Mountain Pico are also plotted in Fig. 7. Recent
measurements by *Lata et al.* (2021) demonstrate better agreement with airborne derived IMF $J_\text{het}$ compared to IMF $J_\text{het}$ derived by China et al. (2017), though, when considering uncertainties, all parameterizations show significant overlap.

**3.3 Deposition ice nucleation kinetics.**

We first analyzed DIN kinetics following CNT expressed using the contact angle, $\theta$ (Knopf et al., 2022; Alpert et al., 2011a; Wang and Knopf, 2011). A smaller $\theta$ value indicates greater ice nucleation propensity while $\theta = 180°$ corresponds to
the case of homogeneous ice nucleation (Knopf et al., 2022; Pruppacher and Klett, 1997; Zobrist et al., 2007). Figure 8a shows $J_\text{het}$ as a function of temperature where FT samples demonstrate DIN at higher temperatures compared to MBL samples. Typical $J_\text{het}$ values are in the range of 1000 cm$^{-2}$ s$^{-1}$. Corresponding $\theta$ values are given in Fig. 8b where, at similar temperatures, greater $J_\text{het}$ values correspond to smaller $\theta$ values. $\theta$ values are distributed around 25°. This $\theta$ value is in the range of $\theta$ values observed for carbonaceous laboratory-generated and field-collected particles while mineral dust particles typically exert
smaller $\theta$ values (Wang et al., 2012a). FT samples are associated with $\theta$ values as low as 15°, implying superior DIN compared to the MBL samples. This may be due to the presence of inorganic species such as mineral dust. Figure 8c depicts derived $\theta$ values as a function of RH$_\text{ice}$. This analysis reflects the fact that DIN on the FT samples is superior to DIN proceeding on the MBL samples. Overall, the derived $\theta$ values follow the suggested DIN parameterization by *Wang and Knopf* (2011).

We also analyzed DIN applying $\Delta a_w$ (Knopf et al., 2022; China et al., 2017) as shown in Fig. 9 generating DIN $J_\text{het}$
and $n_\text{s}$ values for examined MBL and FT samples. With increasing $\Delta a_w$ $J_\text{het}$ and $n_\text{s}$ are continuously increasing. For MBL and FT samples, DIN $J_\text{het}$ ~ 550 cm$^{-2}$ s$^{-1}$ and $n_\text{s}$ ~ 7500 cm$^{-2}$, although ice nucleation occurs at lower $\Delta a_w$ for FT samples indicating greater DIN propensity compared to MBL samples. Hence, compared to the IMF case, a clearer difference between

 

MBL and FT samples is visible. DIN $J_\text{het}$ and $n_\text{s}$ for FT samples display a steeper slope than for MBL samples, having greater $J_\text{het}$ and $n_\text{s}$ values at $\Delta a_w \sim 0.2$. We expressed $J_\text{het}$ and $n_\text{s}$ data for MBL and FT samples by linear regressions following

ABIFM with the fit parameters given in Table 4. We compare these two DIN parameterizations with the one derived at the ground site at ACE-ENA during similar time periods (Knopf et al., 2022). The ground-site derived DIN parameterization falls in between the MBL and FT DIN parameterization, though considering the uncertainties, it is not significantly different compared to the parameterizations based on airborne-collected data. We also plotted a recent DIN parameterization for SSA particles (Alpert et al., 2022). Within stated uncertainties, values of $J_\text{het}$ for SSA particles agree with the newly derived DIN

parameterizations while it more closely represents the MBL samples.

## 4. Atmospheric implications

The results of this study clearly demonstrate that the sampling season, DI events, and sampling altitude, specifically MBL versus FT regions, can impact the makeup of the particle population and as such the INPs and their freezing potential. Wintertime MBL samples impacted by DI events show the greatest contrast in terms of particle population and freezing

characteristics with summertime MBL samples in the absence of DI events. FT samples show greater diversity in the physicochemical makeup of the particle population and, for the most part, demonstrate greater ice nucleation propensity. These findings complicate the designation of INP sources when modeling cloud formation. In the study area, DI events which were most frequent during wintertime, have the greatest impact on the particle and INP population. For a well-mixed MBL as encountered in this study, MBL INP parameterizations derived from ground site measurements agree with the airborne derived

INP parameterizations. However, application of ground-site derived INP parameterizations may not always be suitable for predicting INP number concentrations in the FT, although differences can lie within the uncertainties of those parameterizations.

Most of the identified INPs reflect particle-types that are abundant in the particle population. This study corroborates the ice formation potential of SSA particles (Alpert et al., 2022; McCluskey et al., 2018; DeMott et al., 2016; Cornwell et al.,

2021; Knopf et al., 2022), mineral dust particles (Kanji et al., 2017), and organic particles (Knopf et al., 2018). In the presence of these particle types in the MBL and FT, derived IMF and DIN INP parameterization allow for estimation of INP number concentrations. The developed IMF and DIN parameterizations are based on $\Delta a_w$, which allows for a computationally efficient implementation in cloud and climate models while using the same framework as for homogeneous ice nucleation (Alpert et al., 2022; Knopf and Alpert, 2023). This in turn allows for a complete description of IMF, DIN, and homogeneous ice

nucleation based on the same parameter space, again facilitating computational application.



## 5. Conclusions

The population composition, INP morphology and composition, IMF and DIN conditions and associated freezing kinetics of particles in the marine MBL and FT in the Azores Islands region collected during the ACE-ENA campaign during summer and winter show distinct differences. Micro-spectroscopic single-particle analysis by STXM/NEXAFS indicates that the physicochemical properties of the particle population differ among the summer and winter samples and when DI events prevailed. The latter corroborates the findings by *Tomlin et al.* (2021). These differences are also reflected in the conditions under which IMF and DIN were observed for the examined particles. In general, particles collected in the FT show greater morphological and chemical complexity compared to MBL particles which highlights the importance of long-range transport, chemical transformation, and different particle sources which can impact the local particle population. MBL particles were dominated by inorganic-organic mixtures that reflect the ocean influence with SSA particles having the same composition and absorption spectral features as SSA particles and INPs described in previous studies (Alpert et al., 2022; Knopf et al., 2022). We can conclude that for the Azores Island region, seasonal changes and sampling altitude affect the particle population characteristics and as such the INP sources.

INP identifications using SEM/EDX show that most of the INPs are sea salt, sea salt with sulfate, dust, and carbonaceous particles, all of which are associated with OM. Given that these particle types are also present in the general particle population of the samples, they are not unique, at least at the nanometer-scale resolution of the micro-spectroscopic single particle techniques used. This conclusion is similar to that of other studies that have used micro-spectroscopic single particle analytical techniques to examined particle populations and associated INPs or ice crystal residues (Alpert et al., 2022; Knopf et al., 2022; China et al., 2017; Knopf et al., 2014; Lata et al., 2021; Hiranuma et al., 2013). These observations further emphasize the importance of organic compounds for ice nucleation and suggest the necessity of a better molecular understanding of the species involved and their phase state in response to ambient temperature and humidity.

The particles in both MBL and FT samples displayed different conditions with respect to temperature and $RH_{ice}$ for initiating water uptake, IMF, and DIN. MBL particles that experienced DI showed greater susceptibility for water uptake, did not initiate IMF at ~240 K, and displayed less efficient DIN compared to particles collected in the MBL during summertime when there were no DI events. The summertime FT particles, however, displayed different conditions for initiating water uptake, IMF, and DIN compared to the wintertime samples. FT particle samples, in general, were found to have greater ice nucleation propensity compared to the MBL samples. The reason for these differences may lie in differences in particle composition and chemical transformation during long-range transport (Li and Knopf, 2021; China et al., 2017).

IMF freezing efficiencies, expressed in $J_{het}$ and $n_s$ values were derived following ABIFM. MBL IMF in $J_{het}$ and $n_s$ can be expressed by a log-linear regression and agrees with an IMF ABIFM parameterization derived from ACE-ENA ground site measurements (Knopf et al., 2022). This implies that in the well-mixed MBL encountered during ACE-ENA and when accounting for particle number concentration and surface area, airborne and ground sampling of INPs can produce similar freezing kinetics. DIN analyzed in terms of CNT, derives the contact angle from $J_{het}$ as a function of temperature and $RH_{ice}$



and results followed the DIN parameterization given by *Wang and Knopf* (2011). DIN is also expressed as a function of the
water activity criterion, $\Delta a_w$, resulting in a log-linear regression of $J_{\text{het}}$ and $n_s$ values as a function of $\Delta a_w$. DIN
parameterization for FT particle samples shows greater ice nucleation propensity compared to MBL particle samples.

The differences in the IMF and DIN kinetics analyses reflect the differences in the particle population which, in this
study depended on the seasonal sampling period and sampling height. The unique opportunity to collect particles in the MBL
and FT during different seasons allowed us to examine the corresponding particle populations and INPs. This study emphasizes
the importance of INP measurements that cover seasonal changes and a range of altitudes. For a well-mixed MBL, INP
measurements at ground level and airborne may result in similar data. Hence, for mixed-phase clouds that form within the
well-mixed MBL, which includes certain types of Arctic mixed-phase clouds, ground site INP measurements may suffice for
particle characterization. However, in lower latitudes where the ice melting line is above the MBL, additional sampling of
INPs aloft would be preferential to better establish the INP sources.


**Data availability.** All data needed to draw the conclusions in the present study are given in the paper or in the Supplement.

**Supplement.** The supplement related to this article is available online at:

**Author contributions.** DAK envisioned and supervised the project, performed STXM/NEXAFS experiments and analysis,
conducted ice nucleation analyses, and wrote the first draft of the manuscript. PW conducted ice nucleation experiments and
assisted in STXM/NEXAFS experiments. BW, AL, JMT, RCM, JYA, MAM assisted in STXM/NEXAFS experiments and
analyses. DPV, NNL, SC, conducted SEM/EDX experiments and analyses. JW oversaw aircraft deployment and particle
collection. All authors discussed interpretation of the data and contributed to the writing of the manuscript.


**Competing interests.** Some of the authors are members of the editorial board of Atmospheric Chemistry and Physics. The
peer review process was guided by an independent editor. The authors have no other competing interests to declare.

**Acknowledgements.** This study was supported by the Atmospheric System Research Program and Atmospheric Radiation
Measurement Program sponsored by the U.S. Department of Energy (DOE), Office of Science, Office of Biological and
Environmental Research (OBER), Climate and Environmental Sciences Division (CESD). DAK acknowledges support by the
U.S. DOE grants DE-SC0016370 and DE-SC0021034. RCM and AL acknowledge by the U.S. DOE grant DE-SC0021977.
JW acknowledges funding support from the US DOE grant DE-SC0020259. A portion of this research was performed on a
project award (10.46936/lser.proj.2019.50738/60000088 and 10.46936/sthm.proj.2017.49857/60006200) from the
Environmental Molecular Sciences Laboratory, a DOE Office of Science User Facility sponsored by the Biological and
Environmental Research program under Contract No. DE-AC05-76RL01830. The STXM/NEXAFS particle analysis was
performed at beamlines 5.3.2.2 and 11.0.2 at the Advanced Light Source (ALS) at Lawrence Berkeley National Laboratory.



The work at the ALS was supported by the Director, Office of Science, Office of Basic Energy Sciences, of the U.S. DOE under contract DE-AC02-05CH11231.


**Financial support**. This research has been supported by the U.S. Department of Energy (grant nos. DE-SC0016370, DE-SC0021034, DE-SC0021977, DE-SC0020259, DE-AC05-76RL01830, DE-AC02-05CH11231).

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





**Table 1: Information about collection of particle samples including sample name, substrate type, sampling altitude (above mean sea level), date and time period, and presence of a dry intrusion event.**

| Sample Pair # | | Sampling Altitude (a.m.s.l.) / m | Sampling Date | Sampling Time | Dry Intrusion |
|---|---|---|---|---|---|
| IOP1-MBL1 | Si₃N₄ | 45 ± 6 | 07 Jul 2017 | 1:25:32 PM - 1:35:33 PM | |
| | TEM | 434 ± 174 | | 1:35:33 PM - 1:45:35 PM | |
| IOP2-MBL1 | Si₃N₄ | 89 ± 84 | 25 Jan 2018 | 2:20:59 PM - 2:28:01 PM | yes |
| | TEM | 879 ± 560 | | 2:13:56 PM - 2:20:59 PM | |
| | | 236 ± 11 | | 2:28:01 PM - 2:35:03 PM | |
| IOP2-MBL2 | Si₃N₄ | 40 ± 10 | 01 Feb 2018 | 11:22:10 AM - 11:29:13 AM | yes |
| | TEM | 43 ± 8 | | 1:29:02 PM - 1:36:04 PM | |
| IOP1-FT1 | Si₃N₄ | 1768 ± 120 | 15 Jul 2017 | 1:31:22 PM - 1:41:24 PM | |
| | TEM | 942 ± 119 | | 1:41:24 PM - 1:51:26 PM | |
| IOP2-FT1 | Si₃N₄ | 1586 ± 85 | 25 Jan 2018 | 1:45:45 PM - 1:52:48 PM | yes |
| | TEM | 1447 ± 4 | | 1:38:43 PM - 1:45:45 PM | |
| | | 1643 ± 20 | | 1:52:48 PM - 1:59:51PM | |
| IOP2-FT2 | Si₃N₄ | 893 ± 497 | 30 Jan 2018 | 1:28:24 PM - 1:35:27 PM | |
| | TEM | 1508 ± 2 | | 1:21:21 PM - 1:28:24 PM | |
| IOP2-FT3 | Si₃N₄ | 4055 ± 14 | 19 Feb 2018 | 1:40:35 PM - 1:47:38 PM | yes |
| | TEM | 4057 ± 9 | | 1:33:32 PM - 1:40:35 PM | |






**Table 2: Summary of particle sample information: Sample name, number of particles examined by scanning transmission X-ray microscopy with near-edge X-ray absorption fine structure spectroscopy (STXM/NEXAFS) and computer-controlled scanning electron microscopy (CCSEM) and CCSEM sample surface area examined. CCSEM determined average area equivalent diameter, number of particles examined for ice formation, and estimated particle surface area involved in ice formation experiments. Optical microscopy (OMI) estimated particle surface area involved in ice formation experiments.**


| Sample ID | Number of particles examined by STXM | Number of particles examined by CCSEM | CCSEM Area examined / mm$^2$ | CCSEM Average area equivalent diameter / µm | CCSEM Number of particles examined for ice formation / mm$^2$ | CCSEM Particle surface area estimate for ice formation / cm$^2$ | OMI Particle surface area estimate for ice formation / cm$^2$ |
|---|---|---|---|---|---|---|---|
| IOP1-MBL1 | 352 | 1660 | 0.022 | 0.32±0.18 | 56845±1421 | $(1.21±0.03)×10^{-4}$ | $6×10^{-4}$ |
| IOP2-MBL1 | 241 | - | - | - | - | - | $8.6×10^{-5}$ |
| IOP2-MBL2 | 196 | 966 | 0.080 | 0.22±0.13 | 9160±229 | $(9.50±0.2)×10^{-6}$ | $6.5×10^{-5}$ |
| IOP1-FT1 | 371 | 3633 | 0.022 | 0.22±0.16 | 124409±3110 | $(1.45±0.04)×10^{-4}$ | $1×10^{-4}$ |
| IOP2-FT1 | 121 | 3994 | 0.080 | 0.23±0.16 | 37875±947 | $(4.79±0.1)×10^{-5}$ | $4×10^{-4}$ |
| IOP2-FT2 | 71 | 953 | 0.080 | 0.15±0.09 | 9037±275 | $(4.39±0.1)×10^{-6}$ | $1×10^{-5}$ |
| IOP2-FT3 | 151 | 3160 | 0.218 | 0.19±0.13 | 10992±275 | $(9.47±0.2)×10^{-6}$ | $2.5×10^{-5}$ |




**Table 3. Information about identified ice-nucleating particles (INPs) including temperature and humidity conditions, scanning electron microscopy (SEM) derived area equivalent diameter, and particle-type classification.**

| INP ID | Ice nucleation temperature / K | Ice nucleation humidity $RH_{ice}$ / % | Aera equivalent diameter / µm | Classification |
|---|---|---|---|---|
| IOP1-FT1 | 243 | 135.1 | 1.89 | Dust |
| IOP1-FT1 | 233 | 130.3 | 3.93 | Seasalt |
| IOP1-FT1 | 233 | 130.3 | 1.98 | Seasalt |
| IOP1-FT1 | 223 | 120.4 | 0.65 | Seasalt |
| IOP1-FT1 | 223 | 120.4 | 0.49 | Other |
| IOP1-FT1 | 223 | 120.4 | 0.55 | Seasalt |
| IOP1-FT1 | 223 | 120.4 | 0.73 | Seasalt + Sulfate |
| IOP2-FT1 | 223 | 134.4 | 0.92 | Seasalt + Sulfate |
| IOP2-FT1 | 223 | 134..4 | 0.69 | Seasalt + Sulfate |
| IOP2-FT1 | 223 | 114.3 | 1.02 | Seasalt + Sulfate |
| IOP2-FT3 | 223 | 133.0 | 10.45 | Carbonaceous |
| IOP2-FT3 | 223 | 133.0 | 11.02 | Carbonaceous |





**Table 4. Parameters for derivation of the immersion freezing (IMF) and deposition ice nucleation (DIN) heterogeneous ice nucleation rate coefficient coefficients ($J_{het}$) and ice nucleation active sites (INAS) density ($n_s$) for marine boundary layer (MBL) and free troposphere (FT) particle samples are given as a function of the water activity criterion, $\Delta a_w$,**
**according to $\log J_{het} = c + m \cdot \Delta a_w$ and $\log n_s = c + m \cdot \Delta a_w$. LCL and UCL represent lower and upper confidence levels at 95%, respectively, for the fit parameters. RMSE indicates root mean square error of the fit.**

| Parameterization | $c$ | LCL$_c$ | UCL$_c$ | $m$ | LCL$_m$ | UCL$_m$ | RMSE |
|---|---|---|---|---|---|---|---|
| MBL IMF $J_{het}$ | 1.128 | -0.2636 | 2.52 | 5.791 | 0.5344 | 11.05 | 0.5988 |
| MBL IMF $n_s$ | 2.207 | 0.8156 | 3.599 | 5.791 | 0.5344 | 11.05 | 0.5988 |
| MBL DIN $J_{het}$ | 2.096 | 1.715 | 2.477 | 3.154 | 1.328 | 4.98 | 0.4731 |
| MBL DIN $n_s$ | 3.175 | 2.795 | 3.556 | 3.154 | 1.328 | 4.98 | 0.4731 |
| FT DIN $J_{het}$ | 1.05 | 0.2942 | 1.805 | 12.18 | 7.15 | 17.21 | 0.5706 |
| FT DIN $n_s$ | 2.129 | 1.373 | 2.884 | 12.18 | 7.15 | 17.21 | 0.5706 |









**Figure 1: Representative STXM/NEXAFS analyses of marine boundary layer (MBL) particle samples as indicated by legends. From left to right: false-color particle maps that show particle mixing state composition where OCIN - organic carbonaceous-inorganic, OCEC - organic carbonaceous-elemental carbon, OC - organic carbonaceous. Normalized size-resolved mixing state analysis where IN - inorganic, EC - elemental carbon, OC - organic carbonaceous. Normalized size-resolved organic volume fraction (OVF) per particle where color green intensity represents OC fraction.**







**Figure 2: Representative STXM/NEXAFS analyses of free troposphere (FT) particle samples as indicated. Panels follow description of Fig. 1.**





**Figure 3: Representative STXM/NEXAFS analyses of IOP2-MBL and IOP2-FT samples. Upper panels show false-color particle maps where organic dominated particles are in green (organic mass is > 80%) and inorganic dominated particles are blue (inorganic mass is > 80%). Lower panels show high resolution NEXAFS spectra with indicated organic functional groups including C-C double bonds (C=C), alcohol (C-OH), hydroxyl (COOH), carbonate (CO₃), and inorganic species potassium (K). Corresponding detailed images of particles and their mixing state are given by legend below where IN - inorganic, EC - elemental carbon, OC - organic carbon, and KCO₃ – potassium carbonate.**



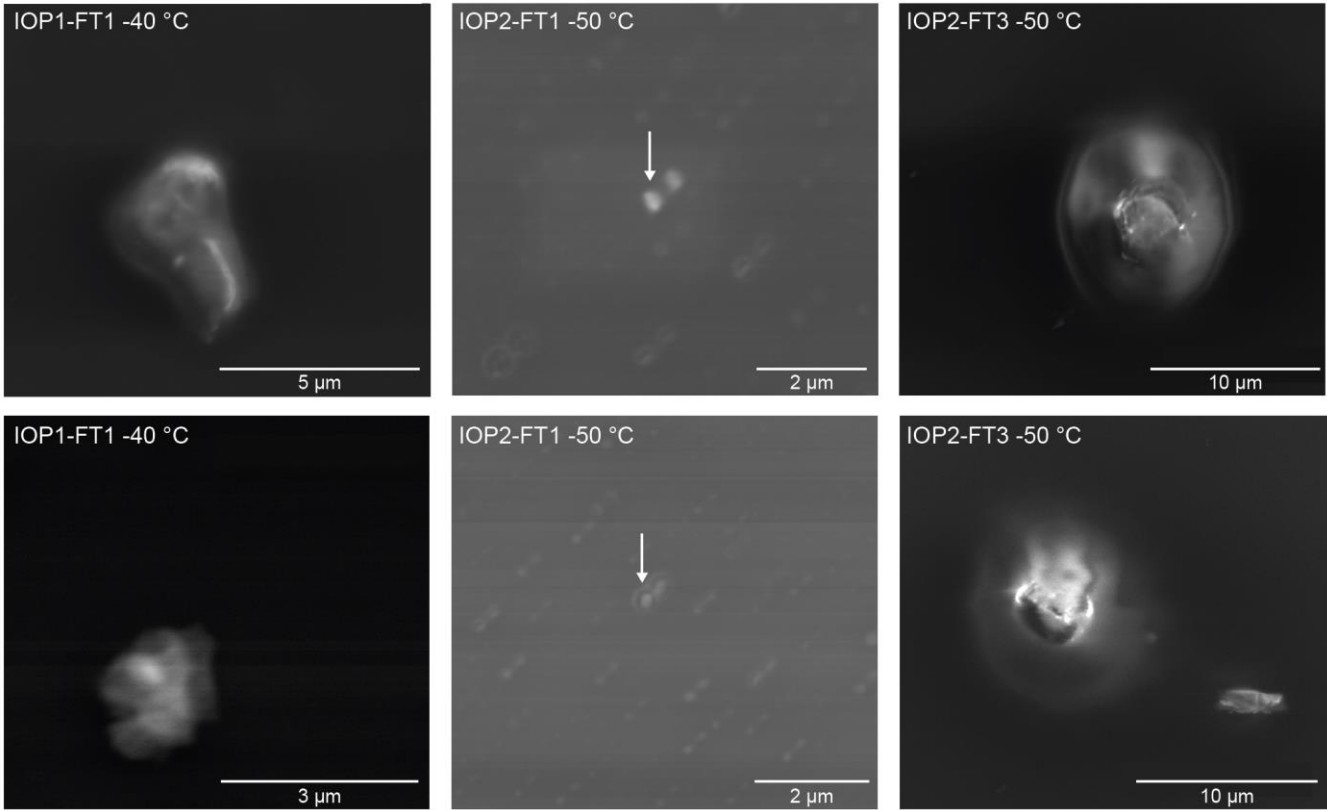

**Figure 4: Identified ice-nucleating particles (INPs) with ice formation temperature indicated. White arrow highlights the INP. The assigned particle types are, starting clockwise from upper left panel: sea salt, sea salt + sulfate, OC, OC, sea salt + sulfate, and sea salt.**









**Figure 5: Composition of experimentally identified INPs. (a) Cumulative atomic percent of elements for 12 identified individual INPs is shown as bars for examined ACE-ENA particle samples. The first column represents the average cumulative atomic percent of elements of the INPs for the specific particle sample. Panel (b) shows representative EDX spectra of identified INPs. \* corresponds to the signal from the substrate (Si$_3$N$_4$-coated silicon wafer chips).**

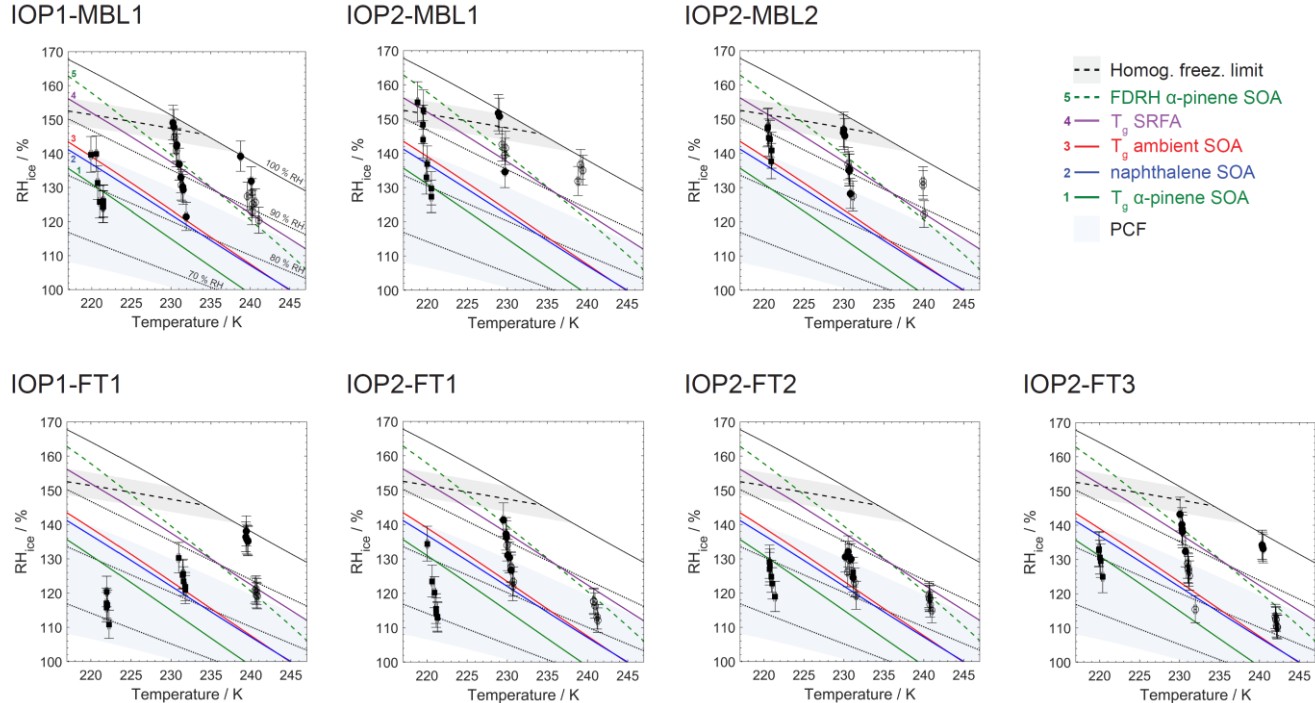

**Figure 6: Ice nucleation and water uptake conditions observed for indicated ACE-ENA particle samples are given as a function of T and RH_ice. IMF - Immersion freezing (solid circles), DIN - deposition ice nucleation (solid squares), water uptake (open circles). The data points and error bars reflect the uncertainties in T and RH_ice. Solid line represents conditions of water saturation (100% RH). Dotted lines indicate constant relative humidity (RH). Dashed line and grey shading represent the homogeneous freezing limit for droplets of 10 μm in size and corresponding uncertainty (Koop, 2004; Koop et al., 2000). The glass transition temperature of laboratory generated α-pinene SOA (green line or 1, (Charnawskas et al., 2017)), naphthalene SOA (blue line or 2, (Charnawskas et al., 2017)) field-derived SOA (red line or 3, (Wang et al., 2012a)), and Suwannee River Fulvic Acid particles (dark violet line or 4, (Wang et al., 2012a)) are plotted. The dashed green line (or 5) displays the full deliquescence relative humidity for α-pinene SOA particles, 500 nm in diameter, under the humidification rate of this experiment (Charnawskas et al., 2017). The light bluish area indicates the conditions for pore condensation freezing (PCF) for pore sizes of 7.5 to 20 nm (Marcolli, 2020, 2014).**




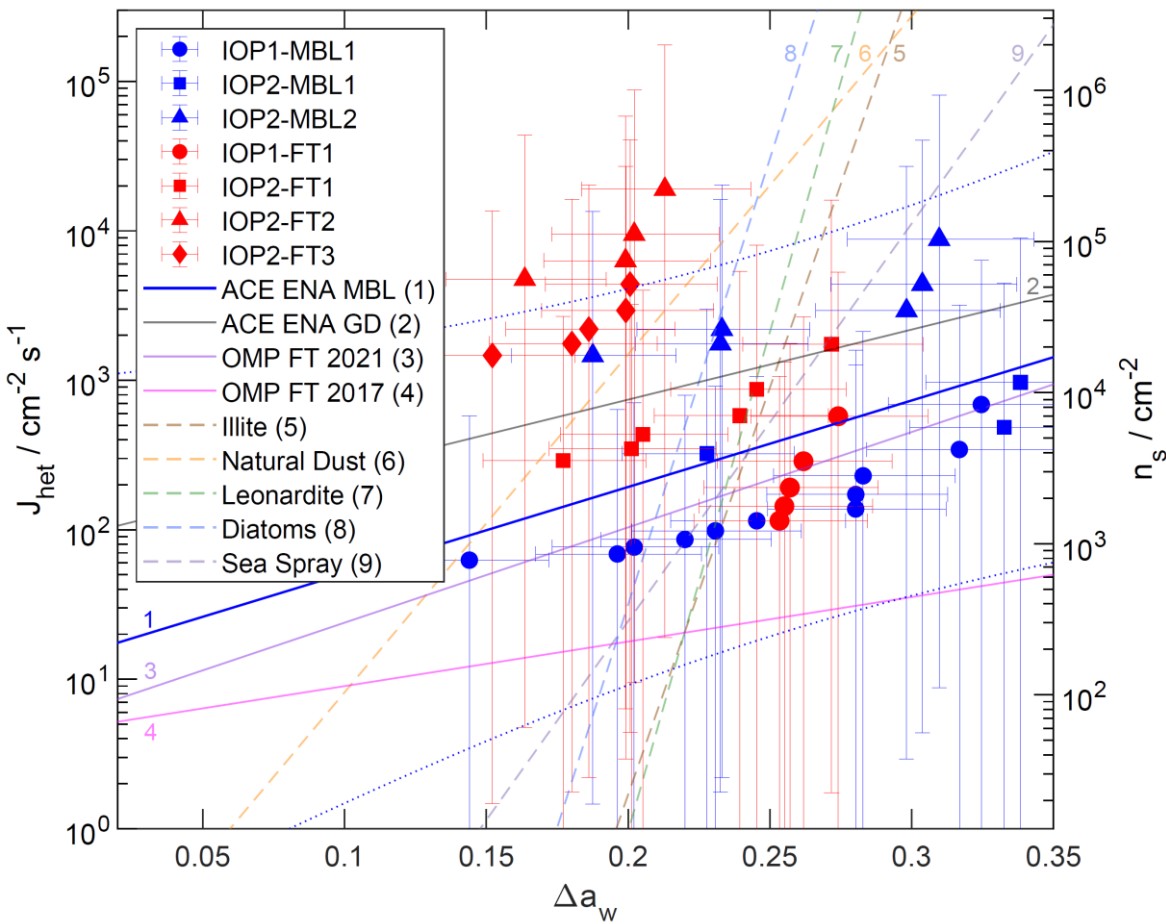

**Figure 7: Immersion freezing (IMF) data of examined ACE-ENA particle samples (solid symbols) and of previous studies (colored lines) as given in legend. Heterogeneous ice nucleation rate coefficients ($J_{het}$) and ice nucleation active sites (INAS) density ($n_s$) are presented as a function of the water activity criterion $\Delta a_w$. Error bars include uncertainties in temperature, humidity, and surface area. Blue solid line represents a linear regression to the newly derived MBL IMF data. Solid black, magenta, and purple lines**

**represent $J_{het}$ and $n_s$ IMF derived from accompanying ACE-ENA ground site INP measurements (ACE-ENA GD), from the Observatory of Mountain Pico (OMP) measurements under free tropospheric (FT) conditions in the Azores on a neighboring island (PMO FT 2017, 2021; (China et al., 2017; Lata et al., 2021)). Please note that only $J_{het}$ was reported for OMP FT 2021 (Lata et al., 2021). Water activity-based IMF $J_{het}$ and $n_s$ for other INP types are given as dashed colored lines as indicated in legend.**






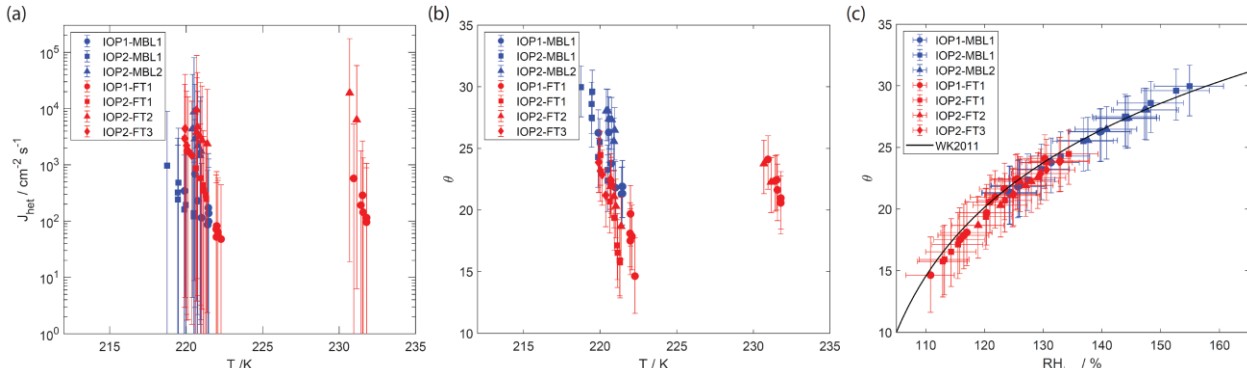

**Figure 8: Deposition ice nucleation (DIN) data of examined ACE-ENA MBL and FT particle samples in blue and red colors, respectively. (a) Heterogeneous ice nucleation rate coefficients ($J_{het}$) as a function of temperature. (b) Contact angles ($\theta$) corresponding to $J_{het}$ values shown in (a). (c) $\theta$ values for relative humidity with respect to ice ($RH_{ice}$) under which DIN was observed. Solid line represents the DIN parameterization by Wang and Knopf (2011).**




Figure 9: Deposition ice nucleation (DIN) data of examined ACE-ENA MBL and FT particle samples (blue and red symbols, respectively). Heterogeneous ice nucleation rate coefficients ($J_{het}$) and ice nucleation active sites (INAS) density ($n_s$) are presented. Error bars included uncertainties in temperature, humidity, and surface area. Blue and red solid and dotted black lines represent linear regression fits and associated fit uncertainties for MBL and FT particle samples, respectively. Black line represents the DIN ABIFM parameterization derived from ground site ACE-ENA INP measurements. Purple line represents the water-activity based DIN sea spray aerosol parameterization from Alpert et al. (2022).