# Peer review of "Physicochemical characterization of free troposphere and marine boundary layer ice-nucleating particles collected by aircraft in the eastern North Atlantic"

_EGUsphere, 2023_

## Author Response (AR1)

Here we provide a detailed point-by-point response to referees #1 and #2. Our responses are given in bold fonts. Line numbers in our response refer to the revised manuscript in which the revisions are highlighted in red fonts.

**Reviewer #1:**

Knopf et al. examine a few aerosol particle samples collected oboard aircraft from within the marine boundary layer and the free troposhere around Azores Islands during different seasons. These samples are throuroughly charaterized with STXM/NEXAFS to derive organic speciation, mixing state and organic volume fraction. They also assessed which particles induce immersion freezing and deposition ice nucleation (and at what thermodynamic condions). The identified INPs were also analyzed for their chemical composition and morpholgy with SEM/EDX.

Overall, the paper is very well written, and I have only mostly minor questions, comments, and editorial suggestions listed below that would further improve the quality of the paper. My recommendation is that this paper undergoes minor revisions and be fully acceptable for publication.

**We thank the reviewer for taking the time to evaluate our manuscript and the general positive note. This detailed review is appreciated and it enhanced the presentation quality of our manuscript.**

The weakest aspect of the paper, in my opinion, is the small number of samples, which of course cannot be changed and is often the reality of aircraft-based sampling, but then the authors draw more general conclusions about the effect of season, altitude etc.. For example, there is only one MBL/FT sample for summer, all other samples are winter, which raises the question about the representativeness of that sample pair for the whole summer in the region. So, without the possibility to collect and analyze more samples, I would wish for a paragraph describing the synoptic situation during sampling and showing that, at least from a meteorological point of view, the sampling conditions are representative for the whole season.

**We agree with the referee's comment. Clearly, we are limited in terms of sampling, a consequence of this type of field measurements. We are also limited in terms of having limited synchrotron-based analytical techniques at hand, i.e., the employed STXM. The first determines the possible number of samples and the second, the number which could be examined. We did not want to give the impression that the samples described represent conditions over entire seasons for all times. We wanted to use them to make the point that we observed differences between seasons and sampling locations which motivate the need for further observations and, e.g., long-term aerosol and INP measurements. To make this clearer we will add in the introduction on line 147 (also in response to referee #2):**

**"Since particle sampling onto substrates for different analyses proceeded in series, not exactly the same air masses and particles could be sampled on the different substrates. Additionally, only a limited number of samples could be examined in detail. For these reasons, the results of this study may not bey valid for all instances of MBL and FT and**

**for entire seasons. However, as demonstrated, the results indicate that there are significant differences in the makeup and ice nucleation propensity among samples collected in the MBL and FT and for different seasons. These new data further motivate the need of long-term studies of INP sources in remote regions such as the Azores."**

L57-59: Can the authors elaborate on what they mean with "insufficient description"?

**In general, models still struggle to reproduce observed ice crystal number concentrations. Our goal here was to point out that INPs are still not sufficiently well characterized, especially when dealing with ambient aerosol particles which are typically more complex in composition. We change the phrase to (line 58):**

**"One possible explanation for this, is the insufficient characterization of the ice nucleating properties of the ambient particles and knowledge of dominant types of INPs in various atmospheric environments (Kanji et al., 2017; Knopf et al., 2018; Cziczo et al., 2017; Knopf and Alpert, 2023)."**

- L61-63: You associate specific aerosol types with specific regions, and while this seems reasonable, I wonder if, for example, North America does not also contribute a significant amount of particles from biomass burning, and if South America is not also a relevant source of anthropogenic emissions?

**Thank you for pointing out some inconsistencies here. Since ACE-ENA is located in the northern hemisphere, the impact of aerosol particles from South America is very likely negligible; in fact, this has not been shown. Also, typical African biomass burning regions are located too far south to be of relevant impact. We will discard the reference Hamilton et al. (2104). Yes, North America contributes biomass burning aerosol to the studied region. Cited Zheng et al. (2020) demonstrate the effect of long range transported biomass burning aerosol particles affecting the composition in the marine boundary layer at ENA. We added further studies that performed backward trajectory analyses and particle composition measurements that point to the impact of North American aerosol particle sources.**

**We change the sentence to (line 62)**

**"Aerosol sources which contribute to the ENA aerosol include natural emissions from the ocean, anthropogenic emissions and biomass burning from North America, and high mineral dust-laden air masses from the Sahara (Wood et al., 2015; Zheng et al., 2020; Alonso-Perez et al., 2012; Zheng et al., 2018; China et al., 2017; Lata et al., 2021)."**

- L72-74: Can the authors describe the DI events a bit more. For example, why are they called dry intrusions in the first place and how much drier are the intruding airmasses in comparison to the surroundings. Also what drives their occurence?

**We refer to the work by Raveh-Rubin (Ilotoviz et al., 2021; Raveh-Rubin, 2017) to provide a response. They are called DI since they originate from the lower stratosphere or upper troposphere where moist content is much lower compared to the planetary boundary layer. This rather dry airmass subsides, resulting in adiabatic warming, further decreasing humidity. See, e.g., Fig. 5 in (Raveh-Rubin, 2017). Exact RH differences cannot be given since this will vary from case to case. However, as shown in (Raveh-**

Rubin, 2017), RH can drop to about 10-20 % RH which then increases again due to mixing with the planetary boundary layer air. In midlatitudes, DIs can be the result of baroclinic wave life cycles and tropopause folds (see review by (Raveh-Rubin, 2017)).

DIs are not the focus of this study. We wanted to provide this additional information, since we have observed in our previous study (Tomlin et al., 2021) that DIs can impact the characteristics of the particle population in the marine boundary layer. This could be another factor possibly explaining differences among particle samples.

We split the original sentence to provide further information (line 74):

"During the wintertime, this region also experiences dry intrusion events (DI) from the FT into the MBL (Ilotoviz et al., 2021; Raveh-Rubin, 2017). Since DIs originate from the lower stratosphere or upper troposphere and adiabatically descend, they are associated with dry air masses that will mix with the MBL moist air (Ilotoviz et al., 2021; Raveh-Rubin, 2017). DIs, in turn, will also transport particles into the MBL thereby impacting the composition of the MBL particle population (Tomlin et al., 2021)."

- L104-105: I find this statement to be far too general. Some mineral INPs are certainly highly efficient, but others have no atmospheric relevance, and the same goes for organic INPs. Also, with INPs it is always important to mention the T/RH regime you are looking at. If the authors want to keep a similar statement in the manuscript, they could follow Kanji et al. (2017), a paper they already reference, and state that mineral dust INPs are more relevant/efficient at temperatures <-20°C, whereas biological INPs are more relevant/efficient at temperatures >-10°C.

Yes, we agree that this is a general statement. We will modify this sentence as suggested (line 106):

"Finally, while mineral dust particles are well known to be some of the most efficient inorganic INPs for temperatures lower than -20 °C (e.g., Murray et al., 2012; Kanji et al., 2017; Cziczo et al., 2017), OM or biological material associated with soil dust particles can also serve as potent INPs active above -10 °C (Kanji et al., 2017; Knopf et al., 2021; Hill et al., 2016; Creamean et al., 2020; Tobo et al., 2014)."

- L130: You could mention how many DI events occured here in the text

We include this information now (line 133):

"We analyzed 7 particle samples collected during 6 different flights including 3 samples from the MBL and 4 from the FT which allows for examination of the role of dynamics (transport and 4 DI events), sampling location, and season in determining the physicochemical properties of the aerosol population and INPs."

- L157-160: The samples for the ice nucleation experiments and for the STXM mesurements are sometimes 100s of meters apart in altitude. Can the authors really ensure that they are representative for each other. For example, for IOP2-MBL1, the TEM grid was sampled to a height of almost 1.5 km, which is so high that I wonder if it can still be considered an MBL sample.

**No, we cannot ensure that those samples are representative. Hence, we provide a range in altitude over which the samples were taken. This is a caveat and not something we can easily improve on. We hopefully communicated this clearly. See also previous comment.**

**Regarding IOP2-MBL1: In response to comments below and to referee #2, we have included boundary layer estimates and min and max sampling altitudes in new Table 1. Table 1 states that 2 TEM grids were employed in sampling particles. The first one was sampling from 34-1891 m and the second one from 215 m to 272 m. The estimated boundary layer height is about 1400 m. As such, we can assume that the majority of sampling occurred within the marine boundary layer.**

Therefore, I would appreciate if the authors could also include vertical T/RH/wind profiles from radio or drop sondes (if available) and derive an average MBL height during the flight for the region. If no direct measurements are available, e.g. ERA5 reanalysis data might be a sufficient replacement.

**We do not have drop sondes information available, but we have potential temperature measurements during the aircraft flights. We have added plots of altitude versus potential temperature (Θ) for each flight date in new Fig. S1. From those we estimate the convective boundary layer height. For flight dates 7/15/17 and 2/19/18 estimation of the boundary height is difficult and associated with greater uncertainties. We added those estimated heights in revised Table 1 and added this additional information to the main text.**

**We added in the supplementary information new Fig. S1:**

[Figure]

**Figure S1. Flight altitude versus potential temperature (Θ) for each flight date as given in Table 1.**

**We revised Table 1 to include estimated marine boundary layer heights and new min and max sampling heights (in response to referee #2):**

Table 1: Information about collection of particle samples including sample name, substrate type, sampling altitude (above mean sea level), estimated marine boundary layer height, date and time period, and presence of a dry intrusion event.

| Sample Pair # | Sampling Altitude (a.m.s.l.) / m | Boundary Layer Height (a.m.s.l.) / m | Sampling Date | Sampling Time | Dry Intrusion |
|---|---|---|---|---|---|
| S-  Si$_3$N$_4$ | Min: 33 | 1600 | 07 Jul 2017 | 1:25:32 PM - 1:35:33 PM | |

| | | | | | | |
|---|---|---|---|---|---|---|
| MBL1 | | Max: 67 | | | | |
| | TEM | Min: 53
Max: 583 | | | 1:35:33 PM - 1:45:35 PM | |
| W-MBL1-DI | Si$_3$N$_4$ | Min: 26
Max: 277 | 1400 | 25 Jan 2018 | 2:20:59 PM - 2:28:01 PM | yes |
| | TEM 1

TEM 2 | Min: 34
Max: 1891
Min: 215
Max: 272 | | | 2:13:56 PM - 2:20:59 PM
2:28:01 PM - 2:35:03 PM | |
| W-MBL2-DI | Si$_3$N$_4$ | Min: 27
Max: 73 | 700 | 01 Feb 2018 | 11:22:10 AM - 11:29:13 AM | yes |
| | TEM | Min: 28
Max: 62 | | | 1:29:02 PM - 1:36:04 PM | |
| S-FT1 | Si$_3$N$_4$ | Min: 1283
Max: 1839 | ~1000 | 15 Jul 2017 | 1:31:22 PM - 1:41:24 PM | |
| | TEM | Min: 663
Max: 1283 | | | 1:41:24 PM - 1:51:26 PM | |
| W-FT1-DI | Si$_3$N$_4$ | Min:1436
Max: 1710 | 1400 | 25 Jan 2018 | 1:45:45 PM - 1:52:48 PM | yes |
| | TEM 1

TEM 2 | Min: 1436
Max: 1455
Min: 1587
Max: 1679 | | | 1:38:43 PM - 1:45:45 PM
1:52:48 PM - 1:59:51PM | |
| W-FT2 | Si$_3$N$_4$ | Min: 73
Max: 1513 | 1100 | 30 Jan 2018 | 1:28:24 PM - 1:35:27 PM | |
| | TEM | Min: 1503
Max: 1512 | | | 1:21:21 PM - 1:28:24 PM | |
| W-FT3-DI | Si$_3$N$_4$ | Min: 4021
Max: 4101 | ~1000 - 3500 | 19 Feb 2018 | 1:40:35 PM - 1:47:38 PM | yes |
| | TEM | Min: 4028
Max: 4076 | | | 1:33:32 PM - 1:40:35 PM | |

**We add on line 174:**

**"Table 1 also provides estimated MBL heights derived from flight altitude versus potential temperature plots given in Fig. S1. Regions in which the potential temperature remains nearly constant are indicative of the convective mixed planetary boundary layer (Stull, 1988). For W-FT3-DI the MBL seems to have been mostly decoupled with varying heights throughout the day. We assigned samples originating from the MBL and FT if sampling of the substrates occurred mostly below and above the estimated MBL height, respectively."**

- ca. L172-173 & Fig S1/S2: Before in L167 the authors write that a DI event is chararcterized by a 400hPa increase withn 24h. 400hPa equates approxiamtely to an airmass coming down from 4200m altitude to the surface. I would have expected to see tat least indications for that in the backtrajectories. Maybe it is just due to teh colorcoding, but I do not see it in the respective figures. Do the authors have an explanation why the DI event is not visible in the backtrajectorie plots?

**Looking at Figs. S1 and S2, we find that the backward trajectory plots reflect descending air masses prior to sampling. One has to keep in mind that the origin of the backward trajectory is not necessarily close to the ground but can be at ~1400 and 4000 m, hence there are differences in airmass descent. However, the general trend of descending air masses is visible in the supplementary figures. We have also changed the scale of the backward trajectory plots as suggested in a later comment.**

**01/25/18: Air masses descent from over 6000 m (green/yellow) to about 2000 m (blue).**

**02/01/18: Air masses descent from over 6000 m (green/orange) to less about 1000 m (dark purple).**

**02/19/18: Some air masses descent from close to 10000 m (orange/red) to about 4000 m (brighter blue).**

- L176: I would suggest that you write out what these qualitatively different trends are in order get your point across more clearly.

**We will discuss the back trajectories for cases IOP2-FT-1 to 3 in more detail, to be more accurate with respect to Fig. S3 (line 190):**

**"For wintertime FT sampling W-FT1-DI and W-FT3-DI, the air masses originated from higher altitudes coinciding with identified DIs. For the case of W-FT2, however, backward trajectories indicate that air masses resided for longer time at similar altitudes. Qualitatively, there seems to be different trends in the air mass trajectories for sampling times during summer and winter. This is mostly due to the presence of midlatitude DIs (Ilotoviz et al., 2021; Raveh-Rubin, 2017). Four out of the 5 winter sampling periods were impacted by DIs where air descended from the upper troposphere towards the sample location."**

- L194: Can the authors describe how the randomness was ensured? Because people have unconscious biases, even when they think they are choosing particles at random.

**To answer this, we need to describe the experimental procedure of analyzing the particles. Ambient particles are deposited on TEM grid substrates of about 2×2 mm$^2$ in size. This grid is made up of about 400 50×50 µm$^2$ square boxes. Numerous particles are deposited across the substrate (can be in the 100000s). Prior to X-ray analysis one cannot resolve all the particles on the substrate. One can get a hint of where the particle loading is likely higher using optical microscopy, however, single particles cannot be visualized except for large accumulations and coarse mode particles. One chooses a "box" on the grid to start the detailed, high resolution, examination. Detected particles are part of the analyzed particle population. Then one moves to the next box. Except for**

trying to be in a substrate area where particle loading is high, there is no bias possible since one cannot "see" the particles before the measurement.

We feel there have been numerous publications on this matter including several reviews by the authors of this study, so that we feel that we do not need to provide such details here.

- section 2.3: Does it ever happen that two or more particles are so close together that they grow too much in the 12s window, making it ambiguous as to which one actually nucleated ice? If so, how do you deal with this and ho often does it occur?

This is a very rare incident and did not occur in this study. It is also a scenario that happens only at higher temperatures (>~ -20 °C) due to the presence of greater water partial pressure that allows for faster ice crystal growth. If this happens, one can still count the ice formation event for freezing kinetic analysis, however, we would not be able to identify the individual particle that served as INP.

- L259-263 & Fig S5 + S6: Is there a reason why you chose monthly-composite satellite products, when your samples were collected only on specfic days? Considering also the backtrajrectories a 10 day composite of the days before sampling might be more relevant for a specific sample? Can you also give average values for the sea surface temperature and chl a concentration (maybe in a 3° window around the Azores to make it consistent with the DIs)?

This text section only serves the purpose of giving an idea why the MS aerosol particle population differs between summer and winter samples. This is, to a degree, speculative due to the limit of samples and our limited knowledge of INPs and their sources. Hence, we placed these figures in the supplement. We mention the general key differences in ocean and atmospheric conditions which might explain differences in particle population:

1. In general, during winter there is greater phytoplankton productivity. This will not change much over a period of several weeks. Same with the trend in ocean temperature. We do not feel that more details are warranted. Since we do not know for sure how marine biological activity, wind speed, and sea spray aerosol (SSA) and associated INPs scale (a matter of research in our group and others), there is no need to perform an analysis for a specific hour or day of sampling which is also beyond the scope of this paper.

2. During winter, near-surface winds are higher (Laurila et al., 2021) and, thus, may result in the generation of more SSA. Hence, in the MBL we would expect MS particle population to be dominated by SSA particles.

3. DIs form during winter along midlatitude storm tracks (Raveh-Rubin, 2017). Those DIs bring air masses and particles into the MBL, independently from ocean conditions and thus could explain changes in MS particle population as shown previously by us (Tomlin et al., 2021). Ocean conditions during which the air mass was sampled at several 1000 m in altitude are very likely not relevant for this discussion.

**To avoid further confusion, we rephrase change the text in this paragraph and add references:**

**We have deleted the original sentence on line 261 starting with "Seasonal change…" and now add on line 280:**

**"We discuss potential factors that might explain the observed different particle population MS among MBL samples collected during different seasons. The Azores are located at the southern border of the North Atlantic storm tracks. While in summer surface winds can be calm, during winter, winds are generally higher (Laurila et al., 2021). Also during winter time, the occurrence of midlatitude DIs coinciding with the North Atlantic storm tracks is much more frequent (Raveh-Rubin, 2017). Ocean conditions such as water temperature and primary production as indicated by chlorophyll *a* concentrations are also different between summer and winter seasons."**

**At the end of this paragraph we add (line 294):**

**"Greater wind speeds and primary productivity during winter may support the presence of larger INOC particles compared to the summer MBL sample while the mixing in of air masses from the upper troposphere from DIs may dilute the local MBL SSA particle population. Clearly, more samples and longer-term studies are needed to address those suggestions."**

- L263-264: The higher wind speeds in winter could be confirmed with e.g. ERA5 reanalysis data.

**Please see the comment above. We cite (Laurila et al., 2021) who provide a 40 year ERA 5 climatological study of North Atlantic surface winds.**

- L283-284: Just from visual inspection of the plots in Fig. 2 I do not clearly see an increase of the inorganic contribution for the DI affected samples. If that is real, what could be an explanation for this difference to Tomlin et al.

**Maybe there has been some confusion when mentioning the study by Tomlin et al. (2021). First, Tomlin et al. referred to relative contributions. Under DI, carbonaceous particles did not decrease in absolute numbers but the contribution of inorganic particles increased, changing the overall fraction of the carbonaceous particles. Tomlin et al. (2021) observed an increase in OVF in particle populations affected by DIs. This is also what we observed, stated on lines 280-282. We clarify this section on line 312:**

**"Based on larger particle samples and application of CCSEM/EDX, Tomlin et al. (2021) observed that the relative carbonaceous contributions to the FT particle population decreases while the inorganic contribution increases during DI events. They also observed an increase in OVF for particle samples impacted by DIs."**

L334: Because not all readers will have read Tomlin et al. (2021), the authors should paraphrase what they refer to here.

**We change the sentence to (line 362):**

**"Furthermore, it highlights the importance of DI events on the MBL particle populations and, potentially, on INP sources. The role of DI events in changing the MS and OVF of the particle populations in the MBL and FT has been demonstrated by Tomlin et al. (2021)."**

- L357-360: I have two points regarding these lines here:

   1. I do not understand to what part of your work you refer to as predictive. It is great work without a doubt, but I did not read it in way that you made preditions about ice nucleation at any point.

   2. In the context of this quote, which essentially says that every marine particle is an INP, any ability to predict becomes a bit meaningless. Which is why I would not include it here, especially if you did show predictive understanding of ice nucleation (see point 1)

**We are not sure that we understand the comments of the reviewer. The two points address very different issues when discussing ice nucleation kinetics.**

**To the first point:**

**From the derived data we perform an ice nucleation kinetics analysis providing two types of parameterizations (CNT/ABIFM and INAS) for two types of ice formation pathways (IMF and DIN) as has been done previously in the literature by several groups. These parameterizations allow to predict the INP numbers when the ambient aerosol particle size distribution and thermodynamic conditions are known. Obviously, the parameterizations are derived from a mixture of different particle types present in the samples. Hence, they are not specific to a certain particle type as, e.g., done in laboratory studies. These parameterizations are suitable for implementation in cloud-resolving models and will provide INP number estimates for this region.**

**Second point:**

**Alpert et al. (2022) provides an ice nucleation description for SSA particles based on classical nucleation theory valid for immersion freezing and deposition ice nucleation. This indeed allows us to predict ice nucleation events from a SSA particle population when SSA size distribution and atmospheric thermodynamic conditions are known.**

**It is not at all meaningless to state that every marine particle *could* serve as an INP. This does not mean that every marine particle becomes activated as INP. Only a small fraction of all marine particles will ultimately "activate", i.e., form an ice crystal, in accordance with the overall nucleation rate (observable ice nucleation events in a given time frame) that is defined by supersaturation and available SSA particle surface area. Keep in mind that Alpert et al. (2022) provide a physical parameterization based on classical nucleation theory to demonstrate that nucleation is stochastic in nature.**

**We have discussed this in several articles and in a recent Nature Reviews Physics article (Knopf and Alpert, 2023; Knopf et al., 2020; Alpert and Knopf, 2016; Knopf et al., 2014; Knopf and Alpert, 2013). Since this is a matter of debate in the community, we also provide an alternative interpretation of ice formation in terms of INAS but this discussion is beyond the scope of this study.**

L386-387: what are the average $J_{het}$ and $n_s$ values for the FT samples?

**We re-arrange this sentence to make it clearer (line 418):**

**"FT samples show a trend of greater $J_{het}$ and $n_s$ values with average $J_{het}$ ~ 700 cm$^{-2}$ s$^{-1}$ and $n_s$ ~ 8000 cm$^{-2}$ compared to MBL samples.**

- L401-402: Can the authors explain briefly why the inclusion of the particle surface area uncertainty produces a steeper slope?

**In short, assuming the same particle type and same size: In experiments some particles induce ice formation at higher and some of the same particles at lower temperatures. One explanation is that, although being of the same material, the particles have different active sites that cause the difference in observed freezing temperature. This is the singular hypothesis or deterministic interpretation. However, we have shown (and in one case proven) that this difference is very likely due to our inability to control particle surface area involved in the commonly applied ice nucleation experiments (Alpert and Knopf, 2016; Knopf and Alpert, 2023; Knopf et al., 2020). We demonstrated that particles that induce ice at higher temperatures are larger than the ones that trigger ice nucleation at lower temperatures. Keep in mind, the "nucleation rate" scales with the inverse of surface area. Hence, if assuming that all particles have the same surface area, at low temperatures the surface area is overestimated ($J_{het}$ or $n_s$ is thus underestimated) and for higher temperatures the surface area is underestimated ($J_{het}$ and $n_s$ is thus overestimated). Correcting for this issue of varying particle surface area, thus, results in a greater increase in ice nucleation efficiency with decreasing temperature, i.e., a change in the slope.**

- L418: Do the 25° refer to only MBL samples or all of them?

**No, only for MBL samples. We rephrase the sentence to (line 450):**

**"$\theta$ values for the MBL samples are distributed around 25°."**

- L427-428: Do the authors have an explanation why the DIN case shows a clearer difference between FT adn MBL samples?

**We do not have a definitive answer to this question, unfortunately. DIN, classically defined, is an ice nucleation mechanism commencing in the absence of liquid water. The particle types present on FT samples are somehow more conducive to initiating ice nucleation compared to MBL samples that are dominated by marine derived particles.**

**We remove "Hence" on line 460 to leave this as a standalone statement.**

- L429: Is there a reason for looking at the values at 0.2? If you look at ca 0.1 and lower, the parametrizations suggest $J_{het}/n_s$ is higher for MBL samples.

**There was no specific reason. The choice of $a_w$=0.2 seems to represent approximately the average of the data set. We can omit this comparison to avoid confusion:**

**Line 461:**

**"DIN $J_{het}$ and $n_s$ for FT samples display a steeper slope than for MBL samples."**

Tab 2 + Fig 7 etc: For sample IOP2-MBL1 no CCSEM data exist. Then, how did you the derive the ice nucleation kinetics? Did you use surface area derived from the optical microscopy instead? If so, this should be mentioned somewhere, especially since the authors themselves correctly mention the importance of an accurate particle surface area (L206).

**Yes, we mentioned this point and caveat on lines 228 – 232:**

**"For analysis of the ice nucleation kinetics, accurate particle surface area is needed. In addition to particle surface area estimates based on optical microscopy, we used particle surface area estimates obtained by computer-controlled SEM/EDX (CCSEM/EDX) (Tomlin et al., 2021) due to its superior resolution compared to optical microscopy (Table 2). When available, these data were acquired from samples collected during the same flights at similar altitudes as the particle samples investigated for this study (Table 1)."**

Editorial:

- I would advise the authors to revise the naming scheme of their samples to somthing that makes that allows easy association with the collection conditions (MBL/FT; summer/winter; DI/no DI). For example, IOP2-MBL1 could become MBL2-summer-DI (2 becasue it's thw 2nd MBL sample in table 1).

  As a reader, I don't know these samples by heart, so I often found myself going back to Tab. 1 to confirm that the sample mentioned in the text is the one I had in mind. This distracts from the actual content. When there was a typo in the sample name, it was even more confusing.

**Referee #2 had a similar comment. We have changed the naming of the samples following both referees' suggestions:**

**IOP1 and IOP2 will be exchanged for "S" and "W" indicating summer and winter, respectively. We include "DI" in the sample name if a DI event was present. E.g.: IOP2-MBL2 will now be W-MBL2-DI.**

- L67, L71, L76, L79: "marine MBL" -> marine marine boundary layer. The double marine is redundant, unless there is some distinction I am unaware of.

**This is erroneous and should just be MBL. This is addressed throughout the manuscript.**

- L103: "inorganic mineral dust" Maybe the authors have a different background, but for me mineral implicitly means inorganic, which would make it redundant here.

**Yes, we can omit "inorganic". We wanted to emphasize the difference to "soil dust" that contains organic matter.**

- L248: "IOP-MBL1" From the context I assume you are referring to IOP**2**-MBL1

**Yes, this is a typo.**

- L274: "IOP-FT0" There is no FT0 sample listed in Tab 1

**Yes, this should be IOP1-FT1, now S-FT1.**

- L362: "IOP-FT1" I assume you refer to IOP**1**-FT1?!

**Corrected.**

- Tab 3.: "Aera equivalent" -> Area

**Typo is corrected.**

- Fig 1: Please use the same colors for the same category in the false color map and the MS plot!

**This has been changed accordingly. Also, for Fig. 2.**

- Fig 3: Please center the text ("Organics coated marine particle") above/below the respective image and ensure that the arrows do not cross the text. Also, why is only the text "Soot" and "K-rich mineral" colored acoording to the respective NEXFS spectra? I would suggest to do either consistently for all particles shown or not at all.

**We centered the text as requested and have all particle descriptions in black font.**

- Fig 4: For the reader's convenience, I would suggest to write the assigned particle type in or above the respective image. The naming is also not consistent with Tab 3: What is called "OC" in Fig. 4, is called "Carbonaceous" in Tab. It is also "Sea salt" in Fig 4 and "Seasalt" in Tab 3. This all should be unified.

**We performed the requested modifications.**

- Fig 6: I have a hard time to distinguish between the different symbols. Maybe different colors would help?

**We have added colors to the symbols to facilitate distinguishing the data sets.**

- Fig 7: I appreciate that the authors show error bars, but find them here almost distracting from the actual data. The authors could consider having a version without error bars in the main manuscript and one with error bars in the SI. The same applies to Fig. 9 and to a lesser to Fig. 8 as well

**We plot Figs. 7-9 without error bars in main text and provide figures with error bars in supplement as Figs. S11-S13.**

- Fig 9: The legend does not match the plot. According to the manucript "ACE ENA GD" should be the line numbered 1 in the plot, but in the legend it is number 3. As a consequence also the legend entry for "ACE ENA MBL" and "ACE ENA FT" does not match the plot.

**This has been corrected.**

- Fig S1 & S2: All maps should have the same scale for the color bar (i.e. the same color should correspond to the same height in all plots). Currently, it is very difficult for the reader to compare the height of the air mass across all plots. The authors should also consider choosing a more color-blind friendly color map.

**We have changed the height scale to a maximum of 10000 m for all plots. We also deleted the countries' states to make the plots more readable.**

- Fig S2: The aspect ratio of the map of sample "IOP2-FT3" got messed up and should be corrected.

**This has been corrected.**

**Reviewer #2:**

Knopf et al. aim at evaluating the ice nucleation properties of the aerosol in the free troposphere and marine boundary layer near the Azores islands, connecting them with the other properties of the aerosol population, using micro-spectroscopic single particle measurement techniques. They also look into the seasonal variations (winter versus summer) and the influence of dry intrusion events. As main findings, they corroborate the season, the altitude, and the occurrence of a dry intrusion affect the physicochemical properties of the particles. Regarding ice nucleation properties, the particles in the free troposphere were more ice active than those collected in the marine boundary layer. However, the seasonal changes in the particle population composition influenced the ability to nucleate ice. Organic matter was present in all samples. They also found that the freezing nucleation rates and INAS density for the boundary layer agree with the parametrizations derived from ground-based measurements in the Azores Islands, verifying the applicability of those parameterizations within the boundary layer. However, this statement doesn't hold for the free troposphere or for latitudes where the temperatures for mix-phase clouds are well above the marine boundary layer.

This paper contributes to characterizing the INP population in areas with low coverage of measurements. However, drawing conclusions about seasonality or the influence of transport dynamics, like dry intrusions, requires a representative pool of samples/data from which one can derive statistically significant trends. Unfortunately, the number of samples analyzed here is limited. The authors should include information that supports the statement of their samples being representative of average conditions for each case (MBL/FT, summer/winter, DI/no DI) in the area under study. Please, consider also the scientific and editorial comments below. After those minor changes, I would support the publication in ACP.

We thank the reviewer for taking the time to evaluate our manuscript and voicing support for publication of our manuscript. Due to the limited number of samples, we did not mean to state that observed data and trends always hold for these time periods in this study area. Our goal was to highlight the fact that distinct measurable differences were observed between MBL and FT and summer and winter times. This makes the case for further studies and long-term measurements to better understand INP sources.

**Scientific**

- L92. One can back up this statement by referring to the BACCHUS database (https://www.bacchus-env.eu/in/) and the latest additions (Welti, A., Thomson, E. S., Schrod, J., Ickes, L., David, R. O., Dong, Z., and Kanji, Z. A.: Overview of ambient ice nucleation measurements from 1949 – 2000, EGU General Assembly 2023, Vienna, Austria, 24–28 Apr 2023, EGU23-1458, https://doi.org/10.5194/egusphere-egu23-1458, 2023.)

**We have added this reference to this statement (line 95).**

- L160. Regarding the information in Table 1 and sample naming, I would suggest changing IOP1 -> S (summer) and IOP2 -> W (winter) and including in the sample name a reference to the occurrence of a dry intrusion or not. Please, clarify what are the ± values that appear in the altitude column. Preferred values would be min and max heights. It is surprising that samples collected at similar altitudes (IOP2-MBL1-TEM and IOP2-FT2-Si3N4) are classified as MBL and FT samples respectively. Information about the boundary layer height would be beneficial.

**We have renamed sample names, also in response to referee #1:**

**IOP1 and IOP2 will be exchanged for "S" and "W" indicating summer and winter. We include "DI" in the sample name if DI event was present. E.g.: IOP2-MBL2 will now be W-MBL2-DI.**

**We have added the boundary layer height in revised Table 1 and included as supplementary figure altitude vs. potential temperature from which the boundary layer height is estimated. Now we present the sampling altitude as min and max heights. During two collection periods fast ascents or descents of the aircraft were involved. Hence, stated sampling altitude range can lie outside MBL or FT definition. However, the great majority of sampling was performed in stated MBL or FT regions.**

**Regarding (IOP2-MBL1-TEM and IOP2-FT2-Si3N4) are classified as MBL and FT samples despite showing similar sampling altitudes: The boundary layer height was different for these two sampling dates.**

**Table 2: Information about collection of particle samples including sample name, substrate type, sampling altitude (above mean sea level), estimated marine boundary layer heigh, date and time period, and presence of a dry intrusion event.**

| Sample Pair # | | Sampling Altitude (a.m.s.l.) / m | Boundary Layer Height (a.m.s.l.) / m | Sampling Date | Sampling Time | Dry Intrusion |
|---|---|---|---|---|---|---|
| S-MBL1 | $Si_3N_4$ | Min: 33 Max: 67 | 1600 | 07 Jul 2017 | 1:25:32 PM - 1:35:33 PM | |
| | TEM | Min: 53 Max: 583 | | | 1:35:33 PM - 1:45:35 PM | |
| W-MBL1-DI | $Si_3N_4$ | Min: 26 Max: 277 | 1400 | 25 Jan 2018 | 2:20:59 PM - 2:28:01 PM | yes |
| | TEM 1 TEM 2 | Min: 34 Max: 1891 Min: 215 Max: 272 | | | 2:13:56 PM - 2:20:59 PM 2:28:01 PM - 2:35:03 PM | |
| W-MBL2-DI | $Si_3N_4$ | Min: 27 Max: 73 | 700 | 01 Feb 2018 | 11:22:10 AM - 11:29:13 AM | yes |
| | TEM | Min: 28 Max: 62 | | | 1:29:02 PM - 1:36:04 PM | |
| S-FT1 | $Si_3N_4$ | Min: 1283 Max: 1839 | ~1000 | 15 Jul 2017 | 1:31:22 PM - 1:41:24 PM | |
| | TEM | Min: 663 Max: 1283 | | | 1:41:24 PM - 1:51:26 PM | |
| W-FT1-DI | $Si_3N_4$ | Min:1436 Max: 1710 | 1400 | 25 Jan 2018 | 1:45:45 PM - 1:52:48 PM | yes |
| | TEM 1 TEM 2 | Min: 1436 Max: 1455 Min: 1587 Max: 1679 | | | 1:38:43 PM - 1:45:45 PM 1:52:48 PM - 1:59:51PM | |
| W-FT2 | $Si_3N_4$ | Min: 73 Max: 1513 | 1100 | 30 Jan 2018 | 1:28:24 PM - 1:35:27 PM | |
| | TEM | Min: 1503 Max: 1512 | | | 1:21:21 PM - 1:28:24 PM | |
| W-FT3-DI | $Si_3N_4$ | Min: 4021 Max: 4101 | ~1000 - 3500 | 19 Feb 2018 | 1:40:35 PM - 1:47:38 PM | yes |
| | TEM | Min: 4028 Max: 4076 | | | 1:33:32 PM - 1:40:35 PM | |

[Figure]

**Figure S1. Flight altitude versus potential temperature (Ө) for each flight date as given in Table 1.**

- L163. How different can the samples be when collected ten minutes apart? Especially for those samples in which the altitude varies significantly during collection (e.g. IOP2-MBL1), what are the measures taken to ensure the comparability of those samples?

**As mentioned earlier, this is a caveat in our sampling ability. By sampling in series using aircraft, one never samples the same air mass. This is true also for instrumentation that samples online. We aimed to sample at similar heights during the same flight to ensure that the different samples are relevant to each other. We hopefully have communicated this caveat sufficiently, but we will add in the introduction on line 147:**

**"Since particle sampling onto substrates for different analyses proceeded in series, not exactly the same air masses and particles could be sampled on the different substrates. Additionally, only a limited number of samples could be examined in detail. For these reasons, the results of this study may not bey valid for all instances of MBL and FT and for entire seasons. However, as demonstrated, the results indicate that there are significant differences in the makeup and ice nucleation propensity among samples collected in the MBL and FT and for different seasons. These new data further motivate the need of long-term studies of INP sources in remote regions such as the Azores."**

- L240. Better contrast for subfigures S3a and b would be appreciated and the inclusion of scale for reference.

**The contrast for subfigures 3a and b has been enhanced and a scale has been added.**

- L263. Please, support with some evidence the statement about stronger winds during wintertime in the ENA region.

**In response to referee #1 who had the same comment, we delete the original sentence on line 276 starting with "Seasonal change…" and now add on line 280:**

**"We discuss potential factors that might explain the observed different particle population MS among MBL samples collected during different seasons. The Azores are located at the southern border of the North Atlantic storm tracks. While in summer surface winds can be calm, during winter, winds are generally higher (Laurila et al., 2021). Also during winter time, the occurrence of midlatitude DIs coinciding with the North Atlantic storm tracks is much more frequent (Raveh-Rubin, 2017). Ocean conditions such as water temperature and primary production as indicated by chlorophyll *a* concentrations are also different between summer and winter seasons."**

- L305. Why were only INPs selected from the FT samples for further analysis? How many particles in total triggered ice formation?

**This was not purposedly decided to only show INPs from FT samples. It is a matter of having high-quality images to be able to relocate the INPs and sufficient instrument time to perform the labor intense work. Figure 6 shows observed ice formation events. We aim to have at least 6 ice nucleation experiments for each investigated temperature, though there are a few instances where we have only 3 repetitive observations. So, there are many more observed ice formation events than the 12 INPs identified and characterized given in Table 3.**

- L327. In my opinion, the error bars accompanying the water uptake measurements from IOP1-MBL1 and IOP2-MBL2 in Figure 6 are comparable. How much lower is the RHice for the IOP1-MBL1 sample?

**Agreed. The range of water uptake with respect to RH$_{ice}$ among the samples is comparable considering the uncertainties. We omit this sentence now (line 357).**

- L448. If most of the identified INPs are particle types abundant in the particle population, what makes some of them ice-active and others not?

**This question has been debated for some time. It comes down to how the process of ice nucleation is interpreted (Knopf and Alpert, 2023; Knopf et al., 2020; Alpert and Knopf, 2016). Classical nucleation theory (CNT) treats nucleation as a stochastic process, thus time dependent, scaling with available surface area and time (see J$_{het}$). To derive a nucleation rate, one has to multiply by particle surface area (all particle with same surface components). The theory then predicts the nucleation events but does not, and in fact, cannot identify the location (or which particle specifically nucleates ice), since it is stochastic in nature. For the given rate and observation time, one particle has to start initiating nucleation. So, no particle is singled out to be more active (assuming all have the same surface properties). In contrast, in a singular hypothesis interpretation (there is no underlying theory but reflects instead a normalization procedure), although particles are the same (same composition, same size, etc.), the activated particle can be identified as having a specific ice-active site. Although the mathematical description does not specify the location of the active site, i.e., which particle will nucleate is actually not known, conceptually, it is thought of as being a "special" particle. For these reasons, we present the analysis with respect to J$_{het}$ (CNT-based) and INAS (ice-nucleation active sites, singular hypothesis description).**

**Editorial**

I would recommend a thorough language check/proofreading to improve both the clarity and reader experience. Previous papers on reporting results from the same campaign (e.g. Knopf et al. (2022)) were easier to follow in my opinion.

**The helpful comments by both referees will enhance the presentation quality of this manuscript accompanied by careful proofreading.**

- L23. Does the abbreviation IOP need to be defined here? In general, make sure that abbreviations are used properly and only defined when necessary. A sentence full of abbreviations makes the manuscript harder to read and follow.

**As suggested above we changed the abbreviation of the samples to make it easier for the reader to follow.**

- L53. This sentence could be split into two for clarity.

**We have changed the sentence to (line 54):**

**"Nevertheless, our predictive understanding of which atmospheric particles act as INPs under conditions typical of mixed-phase clouds in which ice crystals and supercooled**

liquid droplets coexist and cirrus clouds where only ice crystals exist, is still insufficient. Therefore, the implementation of parameterizations for INP number concentration predictions in cloud and climate models remains challenging (Boucher et al., 2013; Storelvmo, 2017; Cesana and Del Genio, 2021; McCoy et al., 2016; Murray and Liu, 2022)."

- L71. MBL and FT are not defined in the main text while other terms (e.g. ENA) are.

**They have been defined in the abstract but not here at their first instance in the main text. We will add the definition on line 73.**

- L88. Split sentence for clarity. MMBL should be MBL. "contribution of inorganic species increased" is redundant information.

**"MMBL" has been corrected.**

**We feel that "contribution of inorganic species increased" is not redundant information. In absolute measures the organic contribution decreased and the inorganic contribution increased as stated in the sentence (line 91):**

**"DI events resulted in a decreased contribution of organic compounds to the population in the MMBL and FT while increasing the contribution of inorganic species."**

**We split the sentence starting on line 92 to:**

**"This in turn has a direct effect on the particles' hygroscopicity and thus CCN properties. Tomlin et al. (2021) concluded that entrainment of particles from long-range continental sources alters the mixing state of the particle population and the CCN properties of aerosol particles in this region."**

- L112. Check for sentences where the same citation is used twice and consolidate. Split sentence for clarity.

**We rephrased this sentence to (line 117):**

**"*China et al.* (2017) demonstrated that most particles were coated by OM and that the identified INPs were mixtures of dust, aged sea salt and soot, and that the OM was acquired either at the source or during transport. Furthermore, they showed that ice formation was promoted by both IMF and DIN pathways."**

- L121. Parts of the sentence appear twice.

**We corrected this sentence (line 125):**

**"To relate the physicochemical properties of the particle population to the identified INPs, Knopf et al. (2022) analyzed particle samples collected at the ACE-ENA ground site during summer 2017."**

- L148. What are the various substrates?

**As outlined below, we have different types of substrates for different analytical techniques. In this instance "various" can be omitted (line 158).**

- L189. Imagining -> imaging

**Corrected (line 208).**

- L192 -194. Please, clarify the example given. It is easier to follow in Thomson (1987).

**Thank you for pointing this out. As we tried to better express the methodology, we realized that we made an error in the calculation of the population uncertainties. We rephrased the explanation of the example and corrected our uncertainty estimates (line 211):**

**"For example, if the aim is to describe the probability of different particle-type classes to be at least 0.95 (or at a significance level of 5%) that all particle-type-class estimates are within 0.05 (or 5 %) of the actual population proportions (i.e., the different particle-type classes on the sample), then a sufficient sample size of randomly chosen particles is 510 particles (Thompson, 1987). This statistical analysis allows us to estimate the uncertainty of the particle population's proportions for any given significance level (here, 5 %) and examined particle number. In this study, sample sizes for STXM analysis varied between 71 to 352 particles, resulting in uncertainties of the particle population proportion (i.e., a particle-type class) of about 13 % and 6 %, respectively. Choosing a lower significance level or having a greater sample size will increase or decrease those uncertainties, respectively. In our case, if, e.g., 30 % of the particles are identified as being organic carbon-inorganic mixtures (OCIN), the uncertainty in how representative that particle-type class is, will be ±6.5 % and ±3 % in examination of 71 and 352 particles, respectively."**

- L220. Change temperature-controlled and refer to temperature-controlled conditions instead.

**We rephrase this sentence for clarity (line 242):**

**"This setup allows one to set the temperature of the substrate, thereby controlling the particle temperature. Inside the cell a humidified flow of ultra-high purity dry nitrogen at 1 standard liter per minute supplies the desired water partial pressure."**

- L274. IOP1-FT0 -> IOP1-FT1

**Corrected: S-FT1 (line 304).**

- L325. Figure 6 shaded areas are not very visible in the printed version of the manuscript. Also, consider choosing different symbols more easily distinguishable.

**Also, in response to referee #1 we have added colored symbols. Printing the pdf version of the figure showed the grey shaded areas clearly. Since the grey shaded areas serve as minor additional information, we would like to leave it and not make it more pronounced to keep the figure easier readable.**

- L360. Can the authors explain what they mean by "predictive understanding"?

**With this phrase we meant that if we know particle composition and we have an ice nucleation parameterization for that specific component, we can predict ice crystal formation. In principle, if the particle's physicochemical properties (morphology,**

composition, mixing state) and component-specific ice nucleation rates are known, we can achieve "closure". This has been outlined and attempted by (Knopf et al., 2021).

**We clarify this statement (line 390):**

**"It also demonstrates that if instrumentation which can resolve physicochemical properties on the nanoscale, e.g., the presence of the surface components, is available in combination with component-specific ice nucleation parameterizations, then atmospheric ice formation can be predicted (Knopf et al., 2021)."**

- L381. The error bars in Figure 7 make it harder to interpret. Consider removing the error bars or presenting two versions. Consider the same in Figure 9.

**Referee #1 had a similar request. We will provide the figure with error bars in the supplement and show in main text the figure without error bars.**

**References**

[revised manuscript text omitted]

variations and key controlling processes, Atmos. Chem. Phys., 18, 17615-17635, 10.5194/acp-18-17615-2018, 2018.

---

## Referee Report (RR1)

I support publication after small technical corrections:

- changing legends and/or captions in figures (e.g. 3, 5, 7, S9, S10. S11, S12, S13) to match the new sample naming scheme.
- double-checking mentions to sample names, like W-MBL2 in line 186 should be W-MBL2-DI.

---

## Author Response (AR2)

We would like to express our gratitude for the careful proofreading of our revised manuscript by the reviewer and editor. Indeed, due to the renaming of all samples and necessary reformatting of almost all figures in the previous round of reviews, we missed several instances of correct naming of samples and correct indication of the presence of a dry intrusion event. We have corrected these instances. Those changes did not impact any of the interpretations or conclusions of the manuscript.

Our responses are given below in bold-faced fonts. Line numbers refer to the manuscript where changes are given in red text font (attached below).

I support publication after small technical corrections:

● changing legends and/or captions in figures (e.g. 3, 5, 7, S9, S10. S11, S12, S13) to match the new sample naming scheme.

**We thank the reviewer for finding instances where the sample naming was still not consistent with the main text. In fact, it also affected figures 1, 6, 8, 9, S2, and S5. The sample names in all figures have been updated as requested.**

● double-checking mentions to sample names, like W-MBL2 in line 186 should be W-MBL2-DI.

**Sample name W-MBL2 is the correct one as used throughout the first part of the manuscript. Then it becomes erroneous. This has been corrected.**

**Please see line 187, we change "These trends coincide…" to "For W-MBL1-DI, this coincides".**

**Line 292: We omit "W-MBL2-DI".**

**Line 357, we change "The S-MBL1 sample shows different ice nucleation propensity compared to the W-MBL-DI samples that were collected during DI events."**

**To "The S-MBL1 sample shows different ice nucleation propensity compared to the W-MBL1-DI and W-MBL2 samples that were collected during a DI event and during winter."**

**Line 359, 360, we change "to W-MBL-DI samples" to "W-MBL1-DI and W-MBL2 samples".**

**Table 1 and 2: We change "W-MBL2-DI"to "W-MBL2" and omit "yes" for dry intrusion.**

[revised manuscript text omitted]